# RESTORE3D: BREATHING LIFE INTO BROKEN OBJECTS WITH SHAPE AND TEXTURE RESTORATION

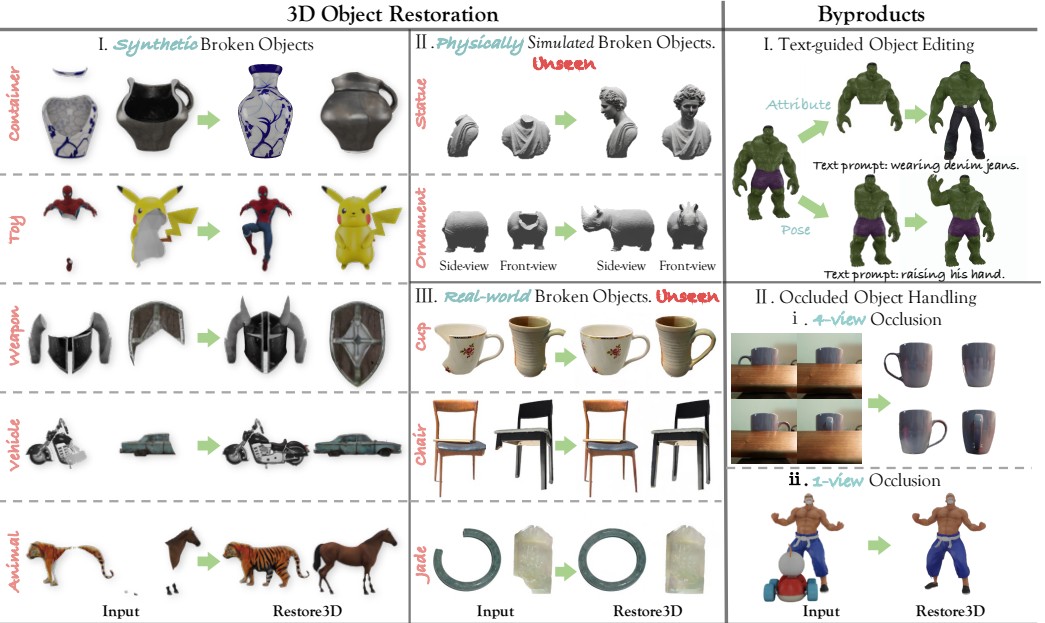

Figure 1: Our **Restore3D** is among the first to simultaneously restore the shape and texture of relatively complex and diverse objects, producing highly plausible and realistic results.

## ABSTRACT

Restoring incomplete or damaged 3D objects is crucial for cultural heritage preservation, occluded object reconstruction, and artistic design. Existing methods primarily focus on geometric completion, often neglecting texture restoration and struggling with relatively complex and diverse objects. We introduce Restore3D, a novel framework that simultaneously restores both the shape and texture of broken objects using multi-view images. To address limited training data, we develop an automated data generation pipeline that synthesizes paired incomplete-complete samples from large-scale 3D datasets. Central to Restore3D is a multi-view model, enhanced by a carefully designed Mask Self-Perceiver module with a Depth-Aware Mask Rectifier. The rectified masks, learned through the self-perceiver, facilitate an image integration and enhancement phase that preserves shape and texture patterns of incomplete objects and mitigates the low-resolution limitations of the base model, yielding high-resolution, semantically coherent, and view-consistent multi-view images. A coarse-to-fine reconstruction strategy is then employed to recover detailed textured 3D meshes from refined multi-view images. Comprehensive experiments show that Restore3D produces visually and geometrically faithful 3D textured meshes, outperforming existing methods and paving the way for more robust 3D object restoration. Project Page: https://iclr-subx.github.io/Restore3D/

## 1 INTRODUCTION

Recent advances in 3D generation and reconstruction techniques (Cheng et al., 2023b; Poole et al., 2022; Lin et al., 2023; Li et al., 2023; Tang et al., 2024) have demonstrated impressive capabil-

Figure 2: **The importance of masks.** In single-view inpainting, user-provided masks define the regions requiring inpainting. However, in a multi-view context, manually creating consistent masks across all views is impractical. Directly inverting object masks to serve as inpainting masks inevitably causes issues (see Prob. 1 & 3). Moreover, manually adjusting masks based on depth information (see Prob. 2) is labor-intensive and time-consuming. As shown in the right figure (a), our mask self-perceiver can automatically indicate the regions that need to be completed. By leveraging both preserved and generated masks (d & e), our approach retains the incomplete object's patterns, ensuring accurate and consistent multi-view inpainting. These masks are also used for the image enhancement stage to yield high-resolution restored images (see Fig. 5).

ities, paving the way for innovative applications across diverse fields. Despite these strides, a significant gap remains in the comprehensive restoration of both shape and texture for broken or incomplete 3D objects. This challenge is particularly relevant for some applications such as cultural heritage preservation, occluded objects reconstruction, and artistic creation, where high-fidelity restoration/completion is crucial.

In this study, we aim to develop a robust framework that can simultaneously restore the shape and texture of incomplete 3D objects while handling complex and diverse data types. Key challenges in achieving this goal include: *i) Data Collection*. Existing 3D datasets (Chang et al., 2015; Dai et al., 2017; Rao et al., 2022) focus primarily on shape completion, often neglecting the equally critical aspect of texture restoration. Furthermore, these datasets typically contain simple objects. Creating a diverse, high-quality dataset remains labor-intensive and time-consuming. *ii) Complexity of Object Completion*. Addressing the intricacies of restoring complex and general objects requires a robust framework, as simpler methods typically work only for limited categories of simple objects, but when applied to more complex cases, they often produce inconsistent or incomplete results. The synthesized regions fail to align with the original parts, or even worse, parts of the original structure are overwritten or discarded during the restoration process. *iii) Consistency Preservation of Broken Parts*. Incomplete objects may exhibit varying degrees of degradation in shape and texture. Therefore, preserving the integrity of original components, including consistent color, style, and structural coherence, is crucial for realistic restoration.

To address these challenges, we propose several complementary solutions: **i) Synthetic Data Generation**. To overcome the limitations of existing datasets, we propose to synthesize paired broken and complete data. **ii) Leveraging Foundation Models**. Recent advancements in foundation models (Hong et al., 2023; Shi et al., 2023; Rombach et al., 2022; Oquab et al., 2023; Kirillov et al., 2023; Yang et al., 2024) have demonstrated exceptional generalizability, due to their extensive architectures, large-scale datasets, and adaptability through fine-tuning. We incorporate foundation models to provide prior knowledge, enabling our framework to effectively handle complex and diverse cases. **iii) Task-Specific Structures**. While foundation models offer valuable priors, task-specific components are necessary to tailor their application. Motivated by studies (Zhang et al., 2023b; Ye et al., 2023; Mou et al., 2023), we guide these models toward optimal probability distributions with specialized modules, achieving more accurate and contextually appropriate restorations.

Concretely, we first produce an automatic pipeline to construct paired data, which uses the Boolean modifier in Blender. It offers diverse and large-scale data that are difficult to acquire manually. Second, we propose an innovative framework named **Restore3D**, comprising two key components, *i.e.*, **multi-view image inpainting and reconstruction**. There are several foundational models (Shi

et al., 2023; Liu et al., 2023a; Xu et al., 2024) in these two components that we can leverage prior knowledge to further handle more diverse incomplete objects effectively. However, simply applying foundational models to multi-view images introduces several **challenges**, as shown in Fig. 2, including: *1) View Inconsistency*: Generated results often differ across views, leading to visual incoherence. *2) Depth Understanding*: Existing models often lack robust depth perception, resulting in failures to recognize occlusions and spatial relationships. *3) Inpainting Position Perception*: Accurately identifying regions requiring inpainting can be difficult, especially for large masks.

To address these issues, we propose a **multi-view** base model combined with a specially designed **mask self-perceiver** module incorporating a **depth-aware mask rectifier**. This module autonomously perceives and reconstructs missing components, preserving the integrity of original broken regions and ensuring consistent results across multiple views. Additionally, by leveraging the preserved and generated masks predicted by the self-perceiver, we can develop an image integration and enhancement pipeline (see Fig. 2 & 5), yielding high-quality and consistent results. To convert high-quality multi-view images into 3D objects, we employ large reconstruction models (LRMs)(Hong et al., 2023; Xu et al., 2024), which offer efficient single- and multi-view object reconstruction capabilities. To overcome the limitation of coarse outputs from these models, we adopt a coarse-to-fine refinement approach. Leveraging recent advances in surface normal prediction models(Bae & Davison, 2024; Ye et al., 2024), we inject normal priors to progressively enhance geometric quality, and refine texture based on updated geometry by using enhanced images. This ensures that our refined shapes and textures maintain high fidelity, even for complex scenarios.

We conduct extensive experiments on Objaverse (Deitke et al., 2023), GSO (Downs et al., 2022), Breaking Bad Dataset (Sellán et al., 2022), Fantastic Breaks (Lamb et al., 2023) and OmniObject3D (Wu et al., 2023) to validate the quality of inpainting and reconstruction. The results demonstrate that our inpainting method significantly outperforms previous approaches (Lugmayr et al., 2022; Zhang et al., 2023b; Rombach et al., 2022), *e.g.*, $\uparrow 13$ in PSNR compared to Nerfiller (Weber et al., 2024). By carefully designing a mask self-perceiver, our method can alleviate view inconsistency, understand depth concepts, and capture inpainting regions, achieving consistent structure and texture styles without requiring user-provided masks to indicate inpainting regions. For reconstruction, our approach enhances both geometric and texture quality as shown in Fig. 1, indicating that our proposed framework is capable of producing complete shapes and textures with relatively high fidelity compared to baseline methods (He & Wang, 2023; Xu et al., 2024; Xiang et al., 2024). Overall, our contributions are summarized as follows,

- To the best of our knowledge, we are among the first to explore the completion of relatively complex shapes and textures. To support this task, we introduce an automated data synthesis pipeline that generates paired incomplete and complete shapes and textures, providing a rich source of training data named RestoreIt-3D.

- We propose Restore3D, a novel framework to tackle shape and texture completion through a combination of multi-view image inpainting and reconstruction. In multi-view image inpainting, we design a mask self-perceiver with a depth-aware mask rectifier for autonomous perception and reconstruction of missing components, ensuring preservation of original features. Moreover, we introduce an image integration and enhancement pipeline to restore fine details. We refine coarse meshes by using normal priors and enhanced images.

- Comprehensive experiments validate the effectiveness of Restore3D, demonstrating its ability to produce complete and high-quality textured meshes.

## 2 RELATED WORK

**2D Inpainting and Generation models** 2D inpainting methods are designed to complete missing content in an image using a given image and mask. LaMa (Suvorov et al., 2021) utilizes fast Fourier convolutions, a large receptive field, and extensive training masks to effectively fill large missing areas, producing plausible inpainting results. Recent advancements in image generation (Rombach et al., 2022; Zhang et al., 2023b) have demonstrated superior performance and can be adapted for inpainting tasks with high-quality outcomes. RePaint (Lugmayr et al., 2022) modifies the diffusion generation process, allowing it to be used for inpainting. NeRFiller (Weber et al., 2024) uses grid priors to make the 2D diffusion model produce more consistent multi-view inpainting results. Instant3Dit (Barda et al., 2025) employs a multi-view inpainting model combined with a large recon-

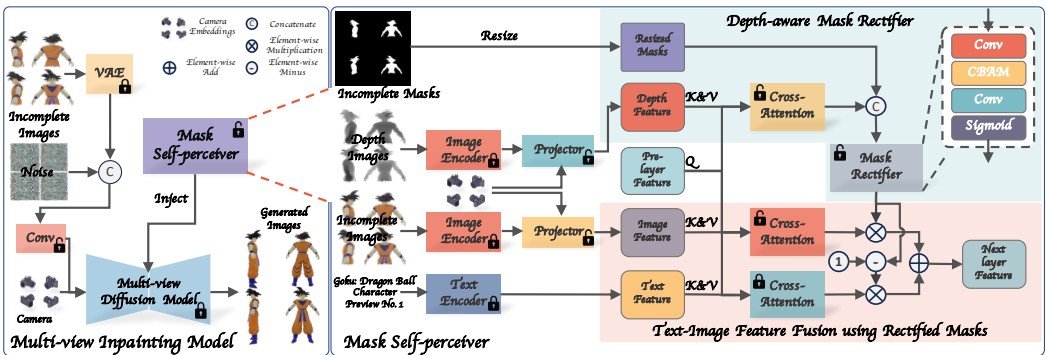

Figure 3: **An overview of multi-view image inpainting.** We carefully design a mask self-perceiver based on a multi-view diffusion model that composes the image and text features with a spatial mask predicted by a depth-aware mask rectifier, therefore the model can automatically perceive the missing part and further generate it meanwhile preserving the original parts.

struction model to enable rapid editing of 3D objects. However, these methods require a user-defined mask to specify the regions that need inpainting.

**3D Generation and Completion** Recent 3D generation models (Wang et al., 2023b; Lin et al., 2023; Chen et al., 2023c) showcase promising results. DreamFusion (Poole et al., 2022) and SJC (Wang et al., 2023a) are first proposed to generate 3D assets from text using the strong 2D text-to-image generation model (Rombach et al., 2022). As 2D diffusion models easily lead to 3D inconsistency, some works (Liu et al., 2023a; Zhou & Tulsiani, 2023; Tang et al., 2023; Szymanowicz et al., 2023; Tewari et al., 2023; Xu et al., 2023) focus on consistent multi-view image diffusion models. MVDream (Shi et al., 2023) uses 3D self-attention and camera embedding to achieve multi-view text-to-image generation. Considering the time-consuming nature of SDS-based methods, there are some works (Face, 2023; Long et al., 2022; Li et al., 2023; Long et al., 2023; Tang et al., 2024; Wu et al., 2024a; Lu et al., 2024) that use multi-view diffusion models and reconstruction models. Another line for 3D generation is that directly train 3D generative models using 3D representations like point cloud (Nichol et al., 2022; Zeng et al., 2022; Luo & Hu, 2021), meshes (Liu et al., 2023b; Gao et al., 2022), neural fields (Kim et al., 2023; Anciukevičius et al., 2023; Müller et al., 2023; Jun & Nichol, 2023; Zhang et al., 2023a; Erkoç et al., 2023; Chen et al., 2023b). In addition to 3D generation, recent 3D shape completion works (Kasten et al., 2023; Zhang et al., 2021; Dai & Nießner, 2019; Mittal et al., 2022; Pan et al., 2021; Cheng et al., 2023b; Chu et al., 2023) usually use different types of 3D representations and networks to model global and local structures, *e.g.*, point cloud, sdf, GAN, VAE, and diffusion models. However, they all learn models on small-scale datasets, therefore the modeling capacity is limited compared with some 3D generation models trained on large-scale datasets (*e.g.*, Objavese (Deitke et al., 2023)). Moreover, these works do not consider the texture.

**Texture Generation.** Several texture generation works (Richardson et al., 2023; Cao et al., 2023; Chen et al., 2023a) use an iteratively texturing strategy based on the pre-trained depth-to-image diffusion models, yielding high-quality texture. However, these methods tend to error lighting inherited from training data. Paint3D (Zeng et al., 2023) proposes a shape-aware UV Inpainting and a shape-aware UVHD diffusion model to alleviate this situation. There is another line to learn texture. Texturify (Siddiqui et al., 2022) employs texture maps on the surface of meshes and uses Style-GAN (Karras et al., 2019) to predict texture. Mesh2Tex (Bokhovkin et al., 2023) incorporates an implicit texture field for texture prediction. These methods are lacking in global information modeling. PointUV (Yu et al., 2023) first trains a diffusion model specifically for mesh texture generation, and the proposed coarse-to-fine framework allows it to enjoy the efficiency of 2D representation while enhancing 3D consistency. Other approaches like AUV-net (Chen et al., 2022), LTG (Yu et al., 2021), and TUVF (Cheng et al., 2023a) learn to generate UV-Maps for 3D shapes. However, they typically focus on the texture generation starting from a complete shape.

## 3 METHOD

### 3.1 DATA PREPARATION & TASK DEFINITION

**Motivation.** We browse the datasets of related tasks and find that the existing datasets (Chang et al., 2015; Deitke et al., 2023; Collins et al., 2022) are not sufficient to handle the shape and texture completion of broken objects, which suggests the need to construct specific broken and complete paired data. However, collecting large-scale paired data in the real world is *time-consuming and labor-intensive*. Thus we propose to *synthesize* broken and complete paired data.

**Data Collection.** We select the recent dataset, G-objaverse (Qiu et al., 2023) that has *more diverse and general objects*, and sample about 83K 3D objects from this dataset.

**Synthesis Pipeiline.** Specifically, we propose an automatic data processing technique using Boolean operations (*i.e.*, Difference and Intersect) of Blender. Additionally, we equip the dataset with text captions using Cap3D (Luo et al., 2023). Subsequently, we normalize and merge the prepared 3D data. The use of Boolean operations requires the introduction of another object. Therefore, we use an ico sphere or cube with random size and rotation angle and then randomly place them inside the 3D bounding box of the prepared 3D data to ensure that the objects can be realistically segmented. After that, it is essential to render this processed data in the format of RGB images to facilitate model learning. We execute the rendering at a resolution of 256×256. The camera settings include a randomly chosen elevation between -10° and 30°. Additionally, the azimuth values are uniformly rendered from 0° to 360° with a randomly sampled start view, producing a total of 32 images per object. The Fov of the camera is randomly from 35° to 45° and the distance is always 2.

**Task Definition.** The 3D object restoration task aims to reconstruct a complete 3D mesh with texture from multi-view images of a damaged object. Given **multi-view images** $\{I_1, I_2, \ldots, I_n\}$ capturing a damaged object from different angles and corresponding **camera parameters** $(K_i, E_i)$, the model will output a complete 3D mesh $M = (V, F, T)$: Vertices, Faces, and Textures.

### 3.2 MULTI-VIEW IMAGE INPAINTING

**Motivation.** Traditional single-view image inpainting methods (Suvorov et al., 2021; Rombach et al., 2022; Zhang et al., 2023b) rely on the user-provided masks that indicate the areas to be inpainted. While this approach works well in the context of single-view images, it presents significant challenges when extended to multi-view contexts as shown in Fig. 2. *1. View inconsistency.* In a multi-view scenario, the user is required to manually provide a mask for each of the views (*e.g.*, four views in our case). This also introduces the risk of errors, as the mask needs to be accurately aligned across different perspectives to maintain 3D consistency. *2. Uncertainty Regarding Inpainting Areas.* These models cannot autonomously perceive the regions that require inpainting when a large mask is applied. Additionally, they do not incorporate depth perception, limiting their understanding of occlusion and spatial relationships. To address these challenges, we propose an innovative approach that enables the model to *ensure view consistency* and *self-perceive the mask*. Concretely, we design the following two parts.

**Mask Self-perceiver.** We propose a mask self-perceiver module based on a multi-view image generation model as shown in Fig. 3. It has two projectors that consist of transformer-based blocks and camera modulation layers, which project the depth and image features $(f_d, f_r)$ extracted from CLIP (Radford et al., 2021) to the diffusion feature space. The camera modulation helps the model to discriminate the feature under different cameras. Then these projected features $(p_d, p_r)$ will be fed to the respective cross-attention blocks as key and value $(\mathbf{K_d}, \mathbf{K_r}, \mathbf{V_d}, \mathbf{V_r})$. The process can be formulated as follows, where $f_*$ can be depth or image features, $p_*$ is the projected features of them.

$$p_* = \mathbf{Proj}(f_*, c) = \mathbf{Trans}(\mathbf{Mod}(f_*, c)) \tag{1}$$

$$s_* = \mathbf{Softmax}(\frac{\mathbf{QK_*^T}}{\sqrt{d}})\mathbf{V}_* \tag{2}$$

Similarly, $s_*$, $\mathbf{K}_*$ and $\mathbf{V}_*$ are the results of $p_*$ via cross-attention and linear layers. $\mathbf{Q}$ originates from the pre-layer features in the diffusion model.

**Depth-aware Mask Rectifier.** Since depth effectively captures the incomplete shape while disregarding texture information, the rectifier can focus solely on identifying the regions that require

Figure 4: **Geometry and Texture Refinement.** We separately refine the geometry and texture of the coarse results inferred by LRMs (Xu et al., 2024).

generation and preservation. Moreover, the depth can help the model understand the spatial relation and occlusion. Specifically, This module leverages depth features obtained after the cross-attention layer, along with incomplete masks, and inputs them into a mask rectifier. The rectifier then outputs a mask indicating where needs to be generated *i.e.*, leveraging the text features and where needs to be preserved *i.e.*, using the image features. The process can be formulated as follows,

$$\mathcal{M}_r = \mathbf{Sigmoid}(\mathbf{Conv}(\mathbf{CBAM}(\mathbf{Conv}[s_d, \mathcal{M}_o]))) \tag{3}$$

$$f_n = (\mathbf{1} - \mathcal{M}_r)s_t + \mathcal{M}_r s_r \tag{4}$$

**Conv** is convolution layers, **CBAM** is Convolutional Block Attention Module (Woo et al., 2018).

**Training objectives** Given training samples, including incomplete images $\mathcal{I}$, depth images $\mathcal{D}$, incomplete masks $\mathcal{M}$, text prompts $\mathcal{P}$ and camera embedding $\mathcal{C}$, the multi-view inpainting loss can be formulated as follows,

$$\mathcal{L} = \min_{\theta} \mathbb{E}_{z, \epsilon \sim \mathcal{N}(\mathbf{0}, \mathbf{I}), t} \| \epsilon - \epsilon_\theta(z_t; t, \mathcal{I}, \mathcal{D}, \mathcal{M}, \mathcal{P}, \mathcal{C}) \|_2^2. \tag{5}$$

### 3.3 IMAGE INTEGRATION AND ENHANCEMENT

**Motivation.** The input resolution of multi-view model is 256 x 256, which is subsequently encoded to 32 x 32 using a Variational Autoencoder. As a result, *local details are compressed, leading to a loss of clarity in both the original and generated regions of the image.* This compression often causes the inpainted part to be unclear, and the reconstructed image may lose fine details that are essential for achieving high-quality results. Moreover, *high-quality images* will *help* the next *reconstruction* stage to give accurate and detailed textured meshes. To address these challenges, we propose a pipeline that enables the model to *restore local details and preserve the original patterns.* **Enhancement Models.** We explore two types of enhancement models. *Real-ESRGAN* (Wang et al.) is effective at preserving the patterns of low-resolution images with minimal misalignment, making it ideal for recovering the overall structure. *ControlNet-Tile* (Zhang et al., 2023b) offers advanced capabilities for enhancing image details, but will modify the original pattern when a high denoising step is used. Based on these properties, we design the following enhancement pipeline. *1. Input resolution alignment using Real-ESRGAN.* Before integrating with the original images, we need to align the resolution. Using Real-ESRGAN effectively preserves the overall structure and does not introduce content that is not related to the original style.

*2. Integration of generated and original parts using rectified masks.* As depicted in Fig. 5, this procedure infers the preserved and generated masks used to compose the images, which preserves the original parts as soon as possible. However, this procedure inevitably leads to some artifacts, *e.g.*, inconsistent color transitions. To address these artifacts, we leverage the mentioned property of ControlNet-Tile to enhance the images. *3. Image harmonizing using ControlNet-Tile with a blending strategy.* Directly using ControlNet-Tile will alter

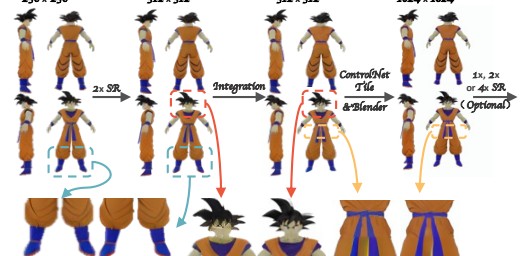

Figure 5: **Image Integration and Enhancement Pipeline using Rectified Masks.**

the original pattern and destroy the integration step. Inspired by previous works (Avrahami et al., 2022; Lugmayr et al., 2022), we incorporate a mask blending technique within the diffusion process. This technique helps maintain the original patterns, eliminates any gaps caused by integration in image space, and enhances the image quality.

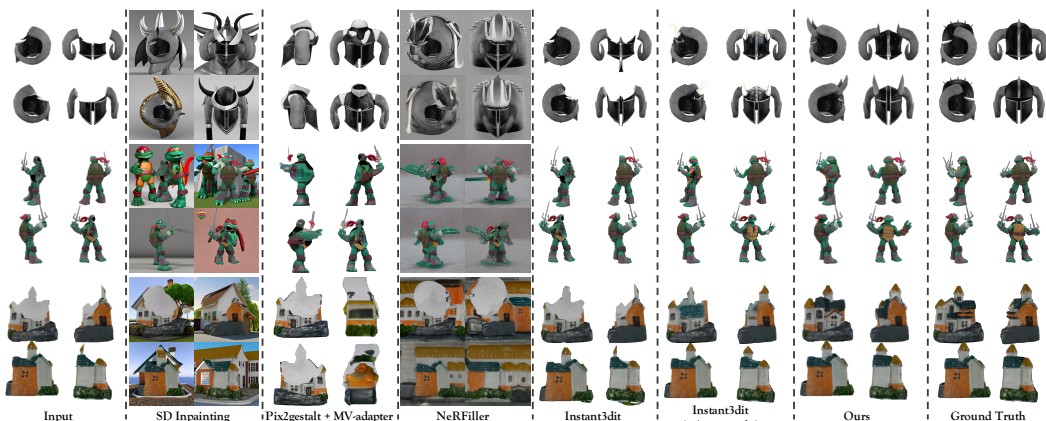

Figure 6: **Visual comparison with inpainting methods.**

## 3.4 MULTI-VIEW IMAGE RECONSTRUCTION

**Fast Reconstruction using Large Reconstruction Models (LRMs).** Recent advancements in LRMs (Hong et al., 2023; Tang et al., 2024; Xu et al., 2024), which leverage sophisticated architectures, large-scale datasets, and extensive model parameters, have demonstrated impressive capabilities in 3D object reconstruction from single or sparse-view images. These models are particularly well-suited for tasks requiring fast mesh reconstruction. However, while LRMs can produce initial reconstructions efficiently, the results are often *coarse and lack the fine details* necessary for high-quality 3D representations. To address this limitation, we adopt a coarse-to-fine schema and refine the shapes and textures of the outputs generated by LRMs, separately, as shown in Fig. 4.

**Geometry Refinement using Normal Prior.** A key component in optimizing shape structure is to obtain high-quality surface normals. Recent surface normal estimation methods (Ye et al., 2024) have demonstrated the ability to predict relatively accurate normals for in-the-wild monocular images or videos. Therefore, we can employ an *off-the-shelf* normal estimation model to provide normal priors and then use it to optimize the shape structure of 3D objects. Since these models are primarily trained on monocular images or videos, the predicted normals are typically in camera space. Thus we need to convert these normals into world space using camera extrinsic parameters. Specifically, we select StableNorm, a model that accepts coarse rendered normals and RGB images as inputs to predict refined normal outputs. The consistency of the rendered normals contributes to the stability and accuracy of the predicted normals, allowing for more precise geometry refinement.

**Texture Refinement using High-quality Images.** Since the current shape differs from the coarse shape, the original texture no longer aligns with the updated geometry. Thus we propose to learn the textures that better match the optimized shape. Concretely, we can use Xatlas to obtain UV coordinates, enabling us to back-project the colors from the inpainted images onto the UV textures. After that, we treat the UV textures as parameters and use the high-quality images to optimize it.

**Training Objectives.** We apply a normal loss $\mathcal{L}_{normal}$ based on the rendered normals $\mathcal{I}_n$ and the target normals $\hat{\mathcal{I}}_n$. Additionally, we apply a mask loss $\mathcal{L}_{mask}$ to ensure that the optimization regions are correctly aligned. The loss function is defined as follows,

$$\mathcal{L}_{shape} = \mathcal{L}_{normal} + \mathcal{L}_{mask} = \|\mathcal{I}_n - \hat{\mathcal{I}}_n\|_2^2 + \|\mathcal{M} - \hat{\mathcal{M}}\|_2^2. \quad (6)$$

To optimize the texture, we use a RGB loss $\mathcal{L}_{rgb}$ on the rendered images $\mathcal{I}_{rgb}$ and enhanced images $\hat{\mathcal{I}_{rgb}}$. The mask loss $\mathcal{L}_{mask}$ is also applied. Moreover, the SSIM $\mathcal{L}_{ssim}$ loss is introduced to improve the texture quality. The loss functions are defined as follows, where $\lambda$ is a weight parameter.

$$\mathcal{L}_{tex} = \mathcal{L}_{rgb} + \mathcal{L}_{mask} + \lambda\mathcal{L}_{ssim} = \|\mathcal{I}_{rgb} - \hat{\mathcal{I}_{rgb}}\|_2^2 + \|\mathcal{M} - \hat{\mathcal{M}}\|_2^2 + \lambda\mathbf{SSIM}(\mathcal{I}, \hat{\mathcal{I}}), \quad (7)$$

## 4 EXPERIMENTS

**Dataset.** For model training, we sample approximately 83K data from the G-objaverse dataset (Qiu et al., 2023) and process them using our proposed pipeline. For model testing, we sample approximately 350 data from the GSO (Downs et al., 2022), Omniobject (Wu et al., 2023), and

Table 1: **Comparison with inpainting and reconstruction methods.** △ means using Depth-Anything (Yang et al., 2024) to obtain the depth images. ♣ means using MV-adapter(Huang et al., 2024). ♡ means using our model's predicted masks as inpainting masks.

(a) **Inpainting.**

| Method | PSNR ↑ | LPIPS ↓ | FID ↓ | SSIM ↑ |
|---|---|---|---|---|
| Repaint | 10.55 | 0.31 | 69.57 | 0.76 |
| SD | 12.58 | 0.22 | 61.15 | 0.83 |
| ControlNet | 10.66 | 0.30 | 69.91 | 0.76 |
| Pix2gestalt ♣ | 16.43 | 0.21 | 75.08 | 0.86 |
| NeRFiller | 12.03 | 0.25 | 65.20 | 0.82 |
| Instant3dit | 19.40 | 0.10 | 48.03 | 0.94 |
| Instant3dit ♡ | 22.37 | 0.07 | 36.08 | 0.95 |
| Ours △ | 25.29 | 0.07 | 32.05 | 0.95 |
| Ours | **25.50** | **0.06** | **31.82** | **0.95** |

(b) **Reconstruction.**

| Method | PSNR ↑ | LPIPS ↓ | CD ↓ | F-Score ↑ |
|---|---|---|---|---|
| Open-LRM | 16.90 | 0.15 | 0.011 | 0.179 |
| InstantMesh | 20.60 | 0.11 | 0.006 | 0.321 |
| Unique3D | 22.00 | 0.14 | 0.005 | 0.306 |
| Direct3D | - | - | 0.006 | 0.297 |
| Trellis | 21.78 | 0.12 | 0.005 | 0.335 |
| Hunyuan3D-2 | 21.31 | 0.14 | 0.006 | 0.346 |
| Amodal3R | 19.37 | 0.15 | 0.008 | 0.248 |
| Ours | **23.35** | **0.09** | **0.005** | **0.389** |

Table 2: **Generalization Ability.**

(a) **Fantastic Breaks Dataset.**

| Method | PSNR ↑ | LPIPS ↓ | SSIM ↑ |
|---|---|---|---|
| SD | 12.59 | 0.72 | 0.40 |
| Controlnet | 15.63 | 0.55 | 0.56 |
| Nerfiller | 18.94 | 0.52 | 0.81 |
| Instant3dit | 23.11 | 0.14 | 0.96 |
| Ours | **26.91** | **0.09** | **0.97** |

(b) **Breaking Bad Dataset.**

| Method | PSNR ↑ | LPIPS ↓ | SSIM ↑ |
|---|---|---|---|
| SD | 12.02 | 0.74 | 0.53 |
| ControlNet | 14.50 | 0.59 | 0.71 |
| NeRFiller | 17.66 | 0.52 | 0.79 |
| Instant3dit | 22.27 | 0.15 | **0.95** |
| Ours | **25.09** | **0.10** | **0.95** |

Objaverse (Deitke et al., 2023) datasets. We also test our model on the Breaking Bad Dataset (Sellán et al., 2022) and Fantastic Breaks (Lamb et al., 2023), which include physically simulated and real-world broken objects, to evaluate its generalizability.

**Metrics.** To assess image quality, we choose Peak Signal-to-Noise Ratio (PSNR), Frechet Inception Distance (FID), Learned Perceptual Image Patch Similarity (LPIPS), and Structural Similarity Index Measure (SSIM). We evaluate geometry quality using Chamfer Distance (CD) and F-scores.

## 4.1 INPAINTING RESULTS.

**Baselines.** We compare our method with single-view image inpainting, *i.e.*, Repaint(Lugmayr et al., 2022), Stable-Diffusion (Rombach et al., 2022), Controlnet (Zhang et al., 2023b), *i.e.*, Pix2gestalt + MV-adater (Ozguroglu et al., 2024; Huang et al., 2024) and multi-view inpainting methods, *i.e.*, Nerfiller (Weber et al., 2024) and Instant3dit (Barda et al., 2025). Note that we do not use the image integration and enhancement pipeline for a fair evaluation.

**Qualitative Comparison.** As shown in Fig. 6, the results demonstrate that our model produces plausible and coherent inpainting outcomes. Previous methods require user-provided masks to guide the model in generating missing parts. When given a relatively large mask, these methods struggle to capture the inherent structure of the objects, leading to less accurate and coherent inpainting. In contrast, our approach does not require predefined inpainting masks. It autonomously perceives and reconstructs missing regions, capturing the underlying structure of the object. This capability allows our method to produce high-quality and structurally consistent inpainting results.

**Quantitative Comparison.** As illustrated in Table 1a, we observe the following: **1)** Our approach achieves the best performance in restoring shape

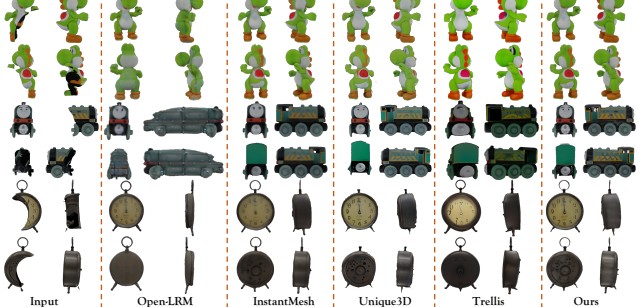

Figure 7: **Visual comparison with reconstruction models.**

and texture. **2)** When applying depth images predicted by Depth-Anything (Yang et al., 2024), our method yields results comparable to those obtained with ground truth depths. **3)** The compared methods produce noticeably inferior results in terms of inpainting quality.

Table 3: **Ablation studies for multi-view inpainting and reconstruction.**

(a) **Inpainting.**

| Method | PSNR ↑ | LPIPS ↓ | SSIM ↑ |
|---|---|---|---|
| IF | 22.65 | 0.14 | 0.90 |
| IF + Conv | 26.53 | 0.08 | 0.94 |
| IF + Conv + DMR | **29.44** | **0.06** | **0.95** |

(b) **Reconstruction.**

| Method | PSNR ↑ | LPIPS ↓ | CD ↓ | F-Score ↑ |
|---|---|---|---|---|
| Baseline | 20.60 | 0.11 | 0.006 | 0.321 |
| GR | - | - | **0.005** | **0.389** |
| GR + TR | **23.35** | **0.09** | **0.005** | **0.389** |

**Generalization Ability.** 1. *Physically simulated broken objects.* As shown in Fig. 1 and Table 2b, we further test our model on the Breaking Bad Dataset (Sellán et al., 2022), synthesized by a physically based method that simulates the natural destruction process of geometric objects. 2. *Real-world broken objects.* As shown in Fig. 1 and Table 2a, we also evaluate our model on Fantastic Breaks (Lamb et al., 2023). These experiments demonstrate the generalization ability of our model to both **unseen** real-world scenarios and physically simulated cases, validating its robustness and practical applicability, despite being trained solely on synthetic data.

## 4.2 RECONSTRUCTION RESULTS.

**Baselines.** We compare our method against both single-view and multi-view LRMs, including LRM (He & Wang, 2023; Hong et al., 2023) and InstantMesh (Xu et al., 2024), Unique3D (Wu et al., 2024a). We also compare our method with image-to-3D generation methods, Direct3D (Wu et al., 2024b), and Trellis (Xiang et al., 2024). For single-view baselines, we input the front-view image. All of the methods use our inpainted and enhanced images as input for a fair comparison.

**Quantitative & Qualitative Comparison.** As shown in Table 1b, our method achieves superior rendered image quality and geometry accuracy, with a substantial improvement over baseline methods. In Fig. 7, it is evident that our approach delivers clearer details and the most accurate geometry among the compared methods. **Training time.** Our approach is highly efficient, requiring 20 seconds per object for geometry and texture refinements.

## 4.3 ABLATION STUDY

**Multi-view Inpainting.** We conduct ablation studies on the proposed multiview Inpainting module in the following components: **1) IF.** Only inputting incomplete images into the cross-attention layers. 2) **Conv.** Concatenating noise and incomplete images to a learnable convolutional layer. 3) **DMR.** Adding the designed Depth-aware Mask Rectifier. As shown in Table 3a, the results improve progressively with each added component, and using all designed components achieves the highest results. As shown in Fig. 8b, 1) IF Only: the model captures the general style of the object but lacks an understanding of spatial relationships and structure. 2) IF + Conv: This enables the model to capture spatial positioning and understand object structure. However, it is

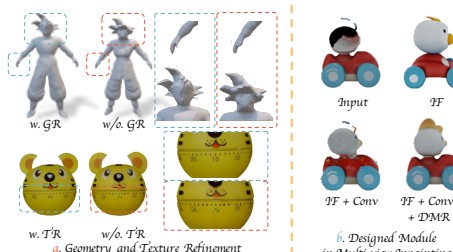

Figure 8: **Visualization of ablation studies.**

still prone to color inaccuracies, especially in areas like the head (blended with error black color). Additionally, the region that needs to be preserved is changed. 3) IF + Conv + DMR: This allows the model to improve its ability to handle occlusions and spatial relationships, producing the best inpainting quality, with coherent colors and well-preserved spatial structure.

**Reconstruction.** We evaluate the impact of the following components: 1) Geometry Refinement (GR), and 2) Texture Refinement (TR). In Table 3b and Fig. 8a, incorporating GR leads to substantial improvements in geometry quality. TR improves the visual quality of rendered images.

## 5 CONCLUSION

In this paper, we propose a novel framework named Restore3D, consisting of multi-view image inpainting and reconstruction, to simultaneously complete both the shape and texture of broken 3D objects. To facilitate this task, we develop an automated data processing pipeline that collects

pair-wise data from a large-scale dataset (Deitke et al., 2023). In the multi-view image inpainting, we design a mask self-perceiver with a depth-aware mask rectifier. This component autonomously identifies and reconstructs missing regions while preserving the original patterns. To address the low resolution resulting from the base model (Shi et al., 2023), we implement an image integration and enhancement pipeline, allowing for seamless integration and detail enhancement by learned masks. For the reconstruction stage, we employ an LRM to quickly generate a coarse result, followed by separate geometry refinement using normal priors and texture refinement using enhanced images. Through this designed framework, our model produces coherent completions of broken objects as illustrated in Fig. 1. Moreover, our designed framework can also handle simple 3D object editing and occluded objects.

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

## A  APPENDIX

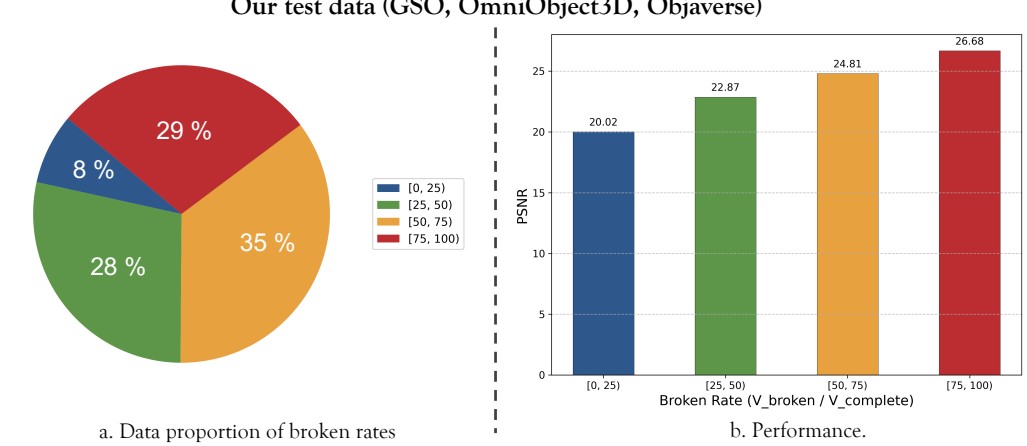

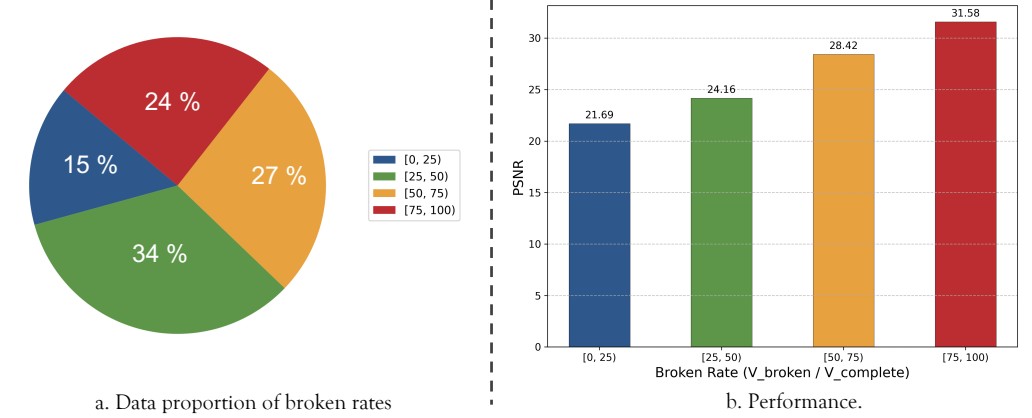

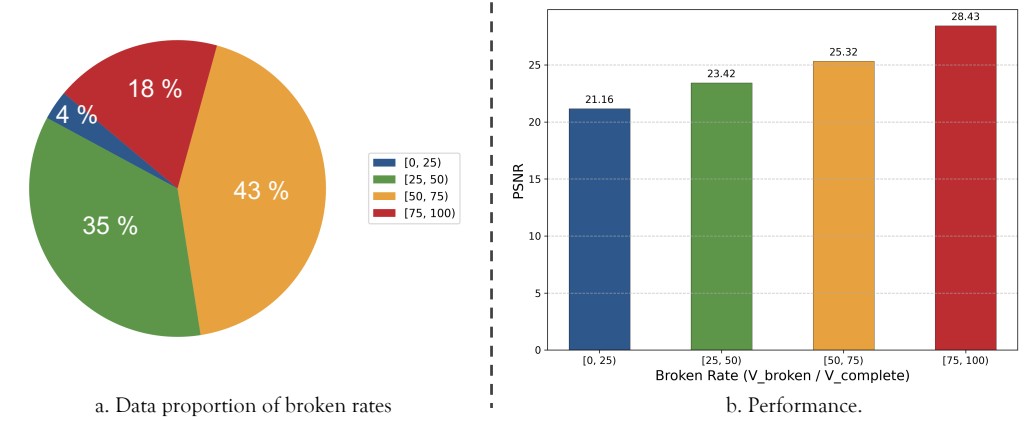

Figure 9: **Broken Rate vs. Performance.**

### A.1  FRAGMENT-SIZE DISTRIBUTION VS. PERFORMANCE

We present the fragment-size distribution and corresponding performance on our synthetic test datasets (GSO, Objaverse, OmniObject3D) and unseen datasets (Breaking Bad Dataset, Fantastic Breaks) in Fig. 9. Across all datasets, performance consistently improves as the missing region be-

comes smaller. This is expected, as smaller missing regions provide the model with richer contextual information, enabling more accurate inference of the missing shape.

Moreover, when the missing region becomes very large, the network gains more flexibility in generating plausible content. In such cases, the output may deviate from the original ground truth, but this discrepancy should not necessarily be considered an "error." This is because extremely large missing regions often provide little or no contextual guidance, leading to inherently ambiguous reconstructions.

## A.2 MORE DETAILS ABOUT IMAGE INTEGRATION AND ENHANCEMENT

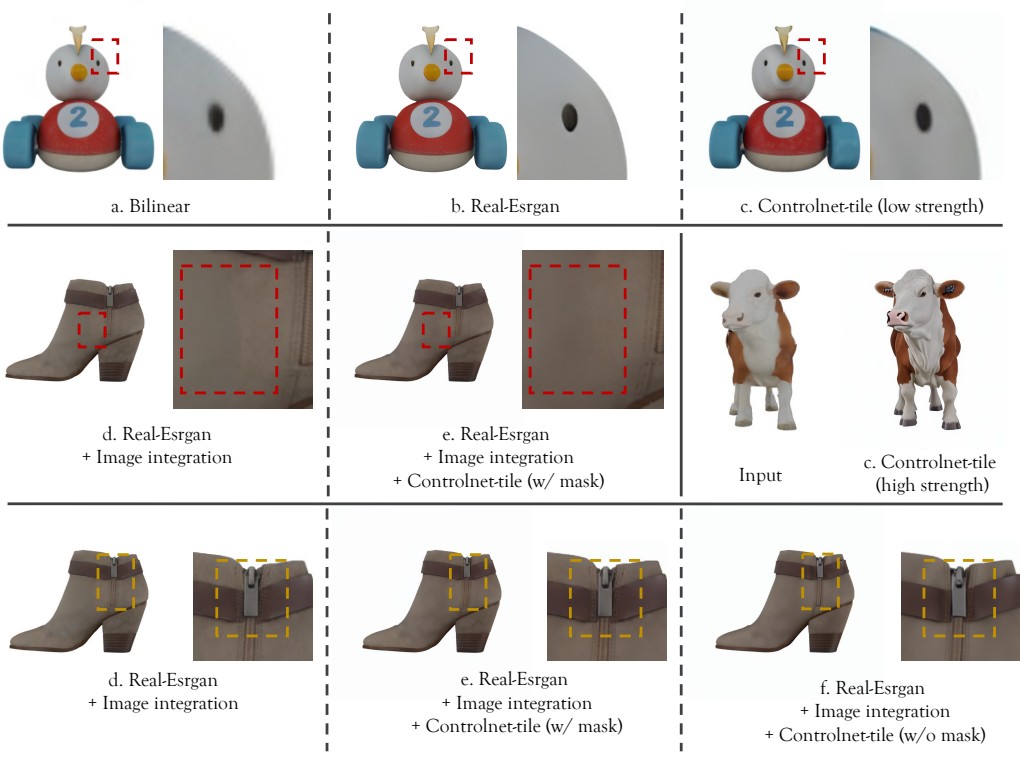

Figure 10: **Visualization of Image Integration and Enhancement**

Table 4: **Ablation studies for Image Integration and Enhancement.**

| Method (256px to 1024px) | PSNR ↑ | LPIPS ↓ | SSIM ↑ |
|---|---|---|---|
| Baseline (Bilinear Upsampling) | 26.83 | 0.10 | 0.97 |
| 4x Real-ESRGAN | 26.59 | 0.08 | 0.97 |
| 4x Controlnet-tile | 26.56 | 0.08 | 0.96 |
| Real-ESRGAN + Image Integration | **27.13** | **0.06** | **0.97** |
| Real-ESRGAN + Image Integration + Controlnet-tile (w/ mask blending) | 26.94 | 0.06 | 0.97 |
| Real-ESRGAN + Image Integration + Controlnet-tile (w/o mask blending) | 26.55 | 0.07 | 0.97 |

We conduct a more detailed ablation study as shown in the Table 4. The visualization results are shown in Fig. 10. We observed that: 1. Solely applying enhancement methods does not improve the quantitative metrics, but can improve visual quality. 2. The performance gains mainly originate from the image integration, which also validates that our rectified mask well indicates the regions requiring inpainting or preservation.

Overall, the organization of this stage is flexible. The key ideas are: 1. Use ControlNet-Tile with a mask-blending strategy to eliminate color inconsistencies during image integration. 2. Upsample the image to the desired resolution using Real-ESRGAN, either before or after the integration step.

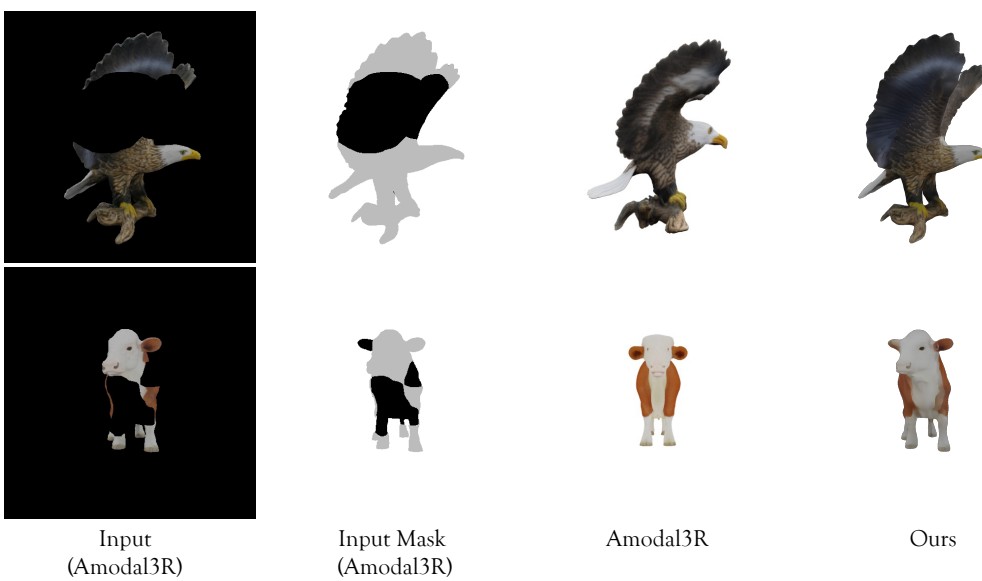

Input          Input Mask         Amodal3R        Ours
(Amodal3R)    (Amodal3R)

Figure 11: **Comparison with Amodal3R.**

### A.3 COMPARISON WITH AMODAL3R.

As shown in Figure 11, We find that the results of Amodal3R often misalign the conditioned images and masks. Furthermore, we notice that the base model used by Amodal3R (Trellis) also faced similar issues with misalignment and inconsistencies, which in turn affected its ability to generate accurate completions.

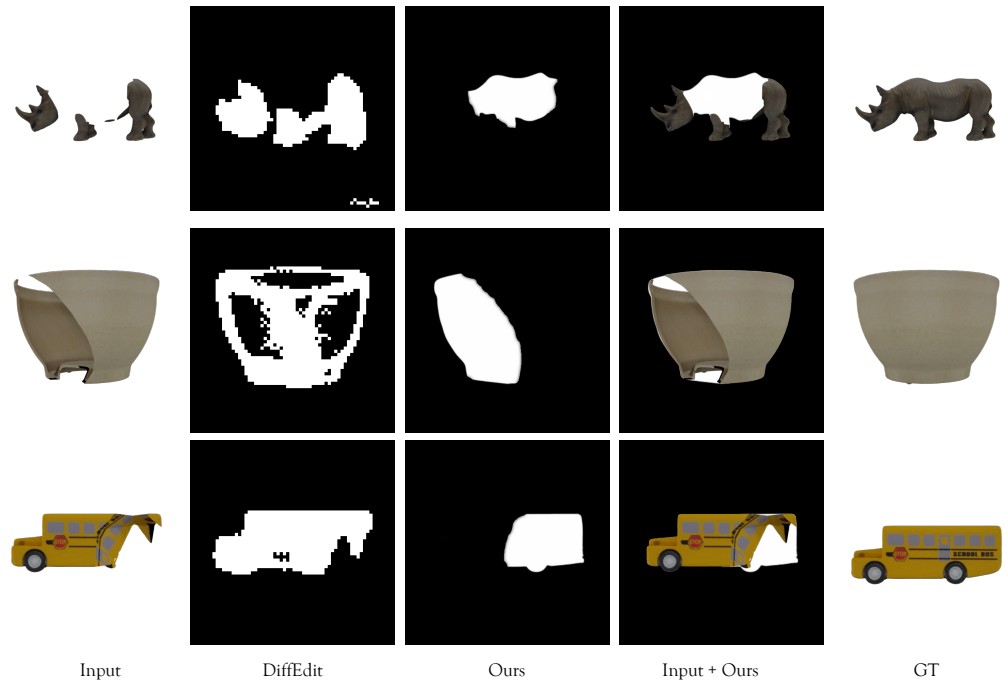

Input         DiffEdit         Ours        Input + Ours        GT

Figure 12: **Comparison with DiffEdit.**

## A.4 COMPARISON WITH DIFFEDIT.

As shown in Figure 12, our method significantly outperforms DiffEdit in terms of mask quality. DiffEdit relies solely on the difference between the noise-conditioned and unconditioned text to infer the mask. However, this approach is suboptimal because it does not account for explicit image and depth information, which are crucial for guiding the model to generate more accurate, contextually appropriate masks in the object restoration task. In contrast, our method incorporates both the image and depth as conditions, significantly improving the quality of mask generation.

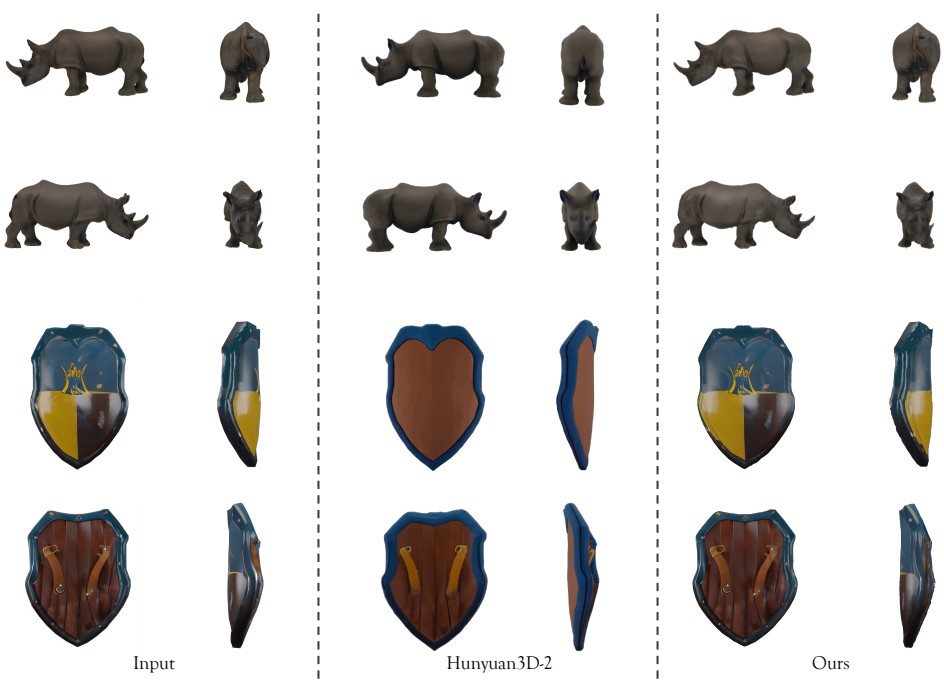

Input           Hunyuan3D-2          Ours

Figure 13: **Comparison with Hunyuan3D-2.**

## A.5 COMPARISON WITH HUNYUAN3D-2.

The shape-generation and texture-generation models in Hunyuan3D-2 are very large, so we use the fast version for inference. Even with the fast version, generating meshes still takes a long time—for example, shape generation alone often requires several minutes, while the texture-generation pipeline typically takes more than 30 minutes. In contrast, our model produces a fully textured mesh in only about one minute. As shown in Figure 13, the results of our model better align with the input images than Hunyuan3D-2.

## A.6 THE ROLE OF COARSE METHES INFERRED BY LRMS

Without LRMs, a typical alternative is to start from a simple primitive (e.g., a sphere) and optimize its shape using our geometry losses. As reported in Table 5, LRMs provide a much better initialization, leading to faster convergence and improved reconstruction quality.

Table 5: **The role of coarse methes inferred by LRMs.**

| Method | CD ↓ | F-score ↑ |
|---|---|---|
| Sphere + geometry optimization | 0.02 | 0.197 |
| LRMs + geometry optimization | 0.005 | 0.389 |

## A.7 COMPUTE BUDGETS.

As shown in Table 6, our model is computationally efficient, runs on modest GPU memory (single NVIDIA RTX 3090 GPU (24GB)), and delivers high-quality results.

Table 6: **Compute budgets.**

| Method | time |
|---|---|
| Inpainting | 5s |
| Integration and enhancement | 13s |
| Coarse mesh reconstruction | 6s |
| Geometry and texture refinement | 20s |
| Total | 44s |

## A.8 VIEW-CONSISTENCY SCORING. & USER PREFERENCES.

As shown in Table 7, we use MEt3R to measure the multiview inpainted images. The table shows that our model outperforms other methods and is very close to the Ground Truth, further validating its effectiveness. We also provide user studies to measure the reconstructed meshes as shown in Table 8. 5 is the best score, 1 is the worst score. The results show our model outperforms other methods.

Table 7: **View-consistency scoring.**

| Method | MEt3R |
|---|---|
| SD | 0.44 |
| Controlnet | 0.53 |
| Nerfiller | 0.50 |
| Pix2gestalt | 0.41 |
| Instant3dit | 0.34 |
| Ours | 0.32 |
| Ground Truth | 0.29 |

Table 8: **User preferences.**

| Method | geometry | texture |
|---|---|---|
| Open-LRM | 1.9 | 2.1 |
| InstantMesh | 3.2 | 3.2 |
| Unique3D | 3.3 | 3.5 |
| Direct3D | 3.0 | - |
| Trellis | 3.1 | 3.0 |
| Hunyuan3D | 3.5 | 3.6 |
| Ours | 3.9 | 4.0 |

## A.9 COMPARISON WITH MVINPAINTER.

We include results for MVInpainter as shown in Table 9. Similar to other baselines, it is unable to accurately perceive the regions that require inpainting. This limitation is reasonable, as MVInpainter is specifically designed for object removal, which is inherently different from our task. Object removal typically involves eliminating an entire object, whereas our task focuses on completing partial regions of an object, leading to fundamentally different requirements and challenges.

## A.10 HYPERPARAMETERS & ROBUSTNESS

Even though our pipeline includes a multi-stage process, our pipeline does not involve a large number of hyperparameters, making it relatively insensitive to hyperparameter choices.

Table 9: **Comparison with MVInpainter.**

| Method | PSNR | LPIPS | SSIM |
|---|---|---|---|
| MVInpainter | 11.12 | 0.29 | 0.79 |
| Ours | 25.50 | 0.06 | 0.95 |

Across all experiments, we do not perform any instance-specific or object-specific hyperparameter tuning; instead, we simply adopt the default or officially recommended settings. This design choice enhances both the practicality and reproducibility of our method. Furthermore, the experimental results in the ablation studies demonstrate that our model can consistently produce plausible outcomes in these standard settings.

The key hyperparameters used in our pipeline are listed in Table 10. A higher Controlnet-tile Strength leads to inconsistent or misaligned results that do not match the preserved (visible) regions. In contrast, moderate Strength values (e.g., 0.25) reliably maintain alignment while enhancing detail quality.

Table 10: **Hyperparameters.**

| Method | Value |
|---|---|
| Multiview inpainting Inference timestep | 50 |
| Multiview inpainting CFG | 5.0 |
| Controlnet-tile Inference timestep | 32 |
| Controlnet-tile CFG | 7.5 |
| Controlnet-tile Strength | 0.25 |
| LRM | official setting |
| StableNormal | official setting |

### A.11 DISCUSSION FOR 3D CONSISTENCY

Our method ensures 3D consistency in the inpainted regions through the following mechanisms:

**Multi-View Constraints (Cross-View Attention).** Our approach enforces 3D consistency by leveraging multi-view constraints, specifically cross-view attention, combined with the strong 3D prior of the base model, MVDream. The MVDream architecture utilizes a block structure that includes both cross-view attention and cross-attention mechanisms. Specifically, our Mask Self-perceiver is applied on the cross-attention layer to aggregate information from the incomplete images and the textual input. This information is then further processed through cross-view attention in subsequent blocks. The inpainting process thus implicitly requires the inpainted content across different views to agree in the latent space, which enforces 3D consistency.

**Geometric Anchors in Incomplete Images.** While the masks do not contain geometric cues, the incomplete images themselves provide essential geometric context. These incomplete images act as geometric anchors, and the diffusion model synthesizes the missing regions in a way that ensures alignment with the visible portions across all views. If the inpainted region were to be inconsistent, it would contradict the visible regions from at least one viewpoint, which the model is trained to avoid such situations.

