# OpenReview forum: "Restore3D: Breathing Life into Broken Objects with Shape and Texture Restoration"
_ICLR.cc/2026/Conference — Submitted to ICLR 2026_

### Official Review · Reviewer_xzb5 · 2025-10-29

**Soundness:** 2
**Presentation:** 3
**Contribution:** 2
**Rating:** 4
**Confidence:** 4

**Summary:**

This paper presents Restore3D, a framework designed to simultaneously recover the shape and texture of fragmented objects from multi-view images. The core component is a multi-view model enhanced by a Mask Self-Perceiver and a Depth-Aware Mask Rectifier. A coarse-to-fine reconstruction strategy is further employed to generate a complete 3D model with detailed textures from the refined multi-view inputs.
Overall, this is a technically comprehensive and systematic work that demonstrates promising results. However, the experimental validation appears limited, comparisons with related 3D reconstruction methods are missing, and ablation studies for key components are not sufficiently conducted.

**Strengths:**

1. Introduce an automated data synthesis pipeline that generates paired incomplete and complete shapes and texture.
2. A technically comprehensive and systematic work for restoring 3D assets that demonstrates promising results.

**Weaknesses:**

1. The mask itself does not provide any 3D geometric constraints, unlike depth or normal maps. How does the method ensure 3D consistency in the inpainted regions when relying solely on the mask?

2. There is a missing comparison with relevant 3D generation and completion baselines (e.g., SD-Fusion), which makes the reported results less convincing.

3. The ablation study is not sufficiently comprehensive. What is the effectiveness of the Masked Self-Perceiver module and the image integration and enhancement process? The overall framework also appears rather complex, which hyperparameters can be tuned, and how sensitive are the results to these settings?

4. Beyond the reported low-resolution limitation, are there other constraints on the method’s ability to complete missing regions? For example, how does the size of the missing area affect reconstruction quality?

**Questions:**

1. The mask itself does not provide any 3D geometric constraints, unlike depth or normal maps. How does the method ensure 3D consistency in the inpainted regions when relying solely on the mask?

2. There is a missing comparison with relevant 3D generation and completion baselines (e.g., SD-Fusion), which makes the reported results less convincing.

3. The ablation study is not sufficiently comprehensive. What is the effectiveness of the Masked Self-Perceiver module and the image integration and enhancement process? The overall framework also appears rather complex, which hyperparameters can be tuned, and how sensitive are the results to these settings?

4. Beyond the reported low-resolution limitation, are there other constraints on the method’s ability to complete missing regions? For example, how does the size of the missing area affect reconstruction quality?

I would revise my rating if the authors solved my concerns

---

> ### Author Response · Authors · 2025-11-26
>
> We sincerely appreciate your recognition of our **automated data synthesis pipeline**, as well as your positive assessment of our work as a **technically comprehensive and systematic solution** for 3D object restoration with promising results. Thank you for your encouraging feedback!
>
> > Q1. 3D consistency
>
> We agree that the mask itself does not contain explicit 3D geometric information. However, our method ensures 3D consistency in the inpainted regions through the following mechanisms:
>
> **• Multi-View Constraints (Cross-View Attention).**
> Our approach enforces 3D consistency by leveraging multi-view constraints, specifically **cross-view attention**, combined with the **strong 3D prior** of the base model, MVDream. The MVDream architecture utilizes a block structure that includes both **cross-view attention and cross-attention** mechanisms.
> Specifically, our **Mask Self-perceiver** is applied on the **cross-attention** layer to **aggregate information** from the incomplete images and the textual input. This **information** is **then** further processed through **cross-view attention** in **subsequent blocks**. The inpainting process thus **implicitly** requires the inpainted content across **different views to agree in the latent space**, which enforces 3D consistency.
>
> **• Geometric Anchors in Incomplete Images.**
> While the masks do not contain geometric cues, the **incomplete images** themselves **provide essential geometric context**. These incomplete images act as **geometric anchors**, and the diffusion model **synthesizes** the **missing regions** in a way that **ensures alignment with the visible portions across all views**. If the inpainted region were **to be inconsistent**, it would **contradict the visible regions** from at least one viewpoint, which the model is **trained to avoid such situations**.
>
> ----------
>
> > Q2. Comparison with SD-Fusion
>
> Thank you for the reviewer’s suggestion. We have included a comparison with SD-Fusion in **Supplementary Materials A.11**. As shown in the results, SD-Fusion consistently produces inferior reconstructions compared to our method.
>
> -----------

---

> ### Author Response · Authors · 2025-11-26
>
> > Q3. 1. The effectiveness of the Masked Self-Perceiver
>
> As shown in **Table 3(a) and Figure 8** of the main paper, the variants **IF only and IF + Conv** correspond to the model **without the Mask Self-Perceiver**. Removing this module significantly **weakens** the model’s ability and results in noticeable degradation. We also further validated its **key internal component**—the depth-aware mask rectifier—which provides additional improvements. These results confirm the necessity of the Masked Self-Perceiver.
>
> > Q3. 2. Image Integration and Enhancement.
>
> Thank you for your insightful suggestion.
> When **comparing our method with other inpainting methods**, we **do not use this pipeline** for a fair evaluation. The pipeline is **only used in the reconstruction stage** to provide **high-quality images** and to serve as the inputs for all methods being compared to ours.
>
> We conduct a more detailed ablation study as shown in the following Table. The **visualization results** are placed in the **A.2 of our paper**.
> We observed that:
> 1. **Solely** applying **enhancement** methods does **not improve** the **quantitative** metrics (as shown in Table b&c) but can **improve visual quality**.
> 2. The performance gains **mainly** originate from the **image integration** (see Table d), which also **validates** that our rectified mask **well indicates** the regions requiring inpainting or preservation.
>
> **• Real-ESRGAN.**
> As mentioned in our paper, Real-ESRGAN is effective at preserving the patterns of low-resolution images with **minimal misalignment** and does **not introduce artifacts** that are inconsistent with the original style.
>
> **• ControlNet-Tile.**
> As mentioned in our paper, ControlNet-Tile offers advanced capabilities for enhancing image details, but it will **modify the original style** when a **high denoising step** is used.
> If used directly **without mask blending** (see Table f), it can **distort the integration** and alter the original pattern.
> **Mask blending** technique helps **preserve** the original patterns and **eliminate** gaps caused by image integration in the pixel space, **improving** the final results (see Table e).
>
> |Method (256px -> 1024px) | PSNR| LPIPS| SSIM|
> |-|-|-|-|
> |**a.** Baseline (Bilinear Upsampling)| 26.83	|0.10	|0.97|
> |**b.** 4x Real-ESRGAN | 26.59	|0.08	|0.97|
> |**c.** 4x Controlnet-tile| 26.56	|0.08	|0.96|
> |**d.** Real-ESRGAN + Image Integration| **27.13**	|**0.06**	| **0.97** |
> |**e.** Real-ESRGAN + Image Integration + Controlnet-tile (**used in our paper**) | 26.94	|**0.06**	|**0.97**|
> |**f.** Real-ESRGAN + Image Integration + Controlnet-tile (w/o mask blending)| 26.55|	0.07 |	**0.97**|
>
> **Overall**, the organization of this stage is flexible. The key ideas are:
>
> 1. Use ControlNet-Tile with a mask-blending strategy to eliminate color inconsistencies during image integration.
>
> 2. Upsample the image to the desired resolution using Real-ESRGAN, either before or after the integration step.
>
>
> > Q3. 3. Hyperparameters
>
> Thank you for the insightful question. We would like to clarify that our pipeline does **not involve a large number of hyperparameters**, and we do **not perform instance-specific or object-specific tuning**. In all experiments, we simply adopt the **regular or official settings**. This design choice contributes to the practicality and reproducibility of our method.
>
> The key hyperparameters used in our pipeline are listed below:
> | Multiview inpainting | Value |
> |-|-|
> | Inference timestep | 50 |
> | CFG | 5.0 |
>
> | Integration and enhanment | Value|
> |-|-|
> | Controlnet-tile Inference timestep | 32 |
> | Controlnet-tile CFG | 7.5 |
> | Controlnet-tile Strength | 0.25 |
>
> A higher Strength leads to inconsistent or misaligned results that do not match the preserved (visible) regions.
> In contrast, moderate Strength values (e.g., 0.25) reliably maintain alignment while enhancing detail quality.
>
> | Geometry and Texture Optimization | Value|
> |-|-|
> | LRM | official setting |
> | StableNormal | official setting |
>
>
> -----------------------------

---

> ### Author Response · Authors · 2025-11-26
>
> > Q4. 1. Large missing area
>
> Indeed, **large missing areas** are the main failure modes as reported in **Supplementary Materials A.5**. We also extend it to three datasets including real-world and physically-simulated datasets in the **A.1 of the main paper**.
>
> 1. Across all datasets, performance **consistently improves** as the **missing** region becomes **smaller**. This is expected, as **smaller missing** regions provide the model with **richer contextual information**, enabling more **accurate inference** of the missing shape.
>
> 2. Moreover, when the **missing** region becomes very **large**, the model inevitably gains **more flexibility** in generating **plausible content**. In such cases, the output may **differ** from the original **ground truth**, but this discrepancy should not necessarily be considered an “error.” This is because extremely **large missing regions** contain **little or no contextual guidance**, leading to inherently **ambiguous** reconstructions.
>
> > Q4. 2. Other limitations
>
> 1) Base Model. The base model may still yield inconsistent results, especially in more complex cases. To address this, using more powerful DiT-based models may provide stronger priors and improve the consistency of the inpainting process.
>
> 2) Texture Realism. Our current method does not fully account for physically-based rendering (PBR) settings, which may affect the realism of the generated textures. To mitigate this limitation, one potential solution is to incorporate material estimation methods to infer materials.

---

> ### Comment · Reviewer_xzb5 · 2025-11-27
>
> The authors have addressed some of my concerns. However, the manuscript still requires major revisions after the rebuttal, so I am keeping my original score.

---

> > ### Author Response · Authors · 2025-11-27
> >
> > Thank you very much for your valuable feedback and for the time you’ve dedicated to reviewing our manuscript.
> >
> > However, we would kindly ask if you could please specify the areas where you feel our revisions have not fully addressed your concerns.
> >
> > We would like to **highlight** that **many of the issues you raised** have **already** been **addressed** in the manuscript **prior to the rebuttal**, including:
> >
> > 1. The effectiveness of the Masked Self-Perceiver – Table 3(a) and Figure 8.
> >
> > 2. Completion baselines (e.g., SD-Fusion) – Supplementary Materials, Appendix A.11.
> >
> > 3. Visualization in Image Integration and Enhancement – Figure 5.
> >
> > 4. Large missing area – Supplementary Materials, Appendix A.5.
> >
> > 5. Limitations – Supplementary Materials, Appendix A.12.

---

### Official Review · Reviewer_aMfZ · 2025-10-29

**Soundness:** 3
**Presentation:** 3
**Contribution:** 3
**Rating:** 6
**Confidence:** 4

**Summary:**

This paper proposes *Restore3D*, a pipeline that simultaneously restores the shape and texture of broken 3D objects from multi-view images. The authors build an automated data synthesis pipeline to generate paired broken/complete data, design a mask self-perceiver with a depth-aware mask rectifier to solve multi-view inpainting without manually provided masks, and further refine geometry and texture using normal priors and image enhancement. Experiments demonstrate clear improvements over baselines in both inpainting quality and reconstructed mesh fidelity.

**Strengths:**

1. The proposed mask self-perceiver with depth-aware mask rectifier is well-motivated and eliminates the need for manually-defined masks in multi-view settings.
2. The experimental results show substantial improvements over existing baselines across multiple datasets, including real and physically simulated broken objects.
3. The paper is overall well-written and should be easy to follow.

**Weaknesses:**

1. The overall pipeline consists of many sequential components (e.g., multi-view inpainting, mask rectification, image integration, and refinement), and this multi-stage dependency may raise concerns regarding robustness to failures in intermediate stages. In addition, it would be helpful to provide a runtime breakdown for each stage to better assess the practicality and deployment cost of the system.
2. The paper would benefit from a comparison against **Amodal3R**, a Trellis-based method for object completion under occlusions. In this setting, the occluded regions predicted by the proposed mask self-perceiver could naturally serve as the amodal masks required by Amodal3R.

[1] Amodal3R: Amodal 3D Reconstruction from Occluded 2D Images. (ICCV25)

**Questions:**

How does the method behave in cases where the broken region is extremely large (e.g., >60% missing)?

---

> ### Author Response · Authors · 2025-11-26
>
> Reviewer aMfZ
>
> We sincerely appreciate your recognition of **our mask self-perceiver with depth-aware rectification**, the strong **experimental** gains across diverse **real and simulated datasets**, and the **clarity of our presentation**. Your positive and encouraging feedback is truly valuable to us. Thank you for your valuable feedback!
>
> > W1:  1. Robustness
>
> Even though our pipeline includes a multi-stage process, our pipeline does **not involve a large number of hyperparameters**, making it relatively **insensitive to hyperparameter choices**.
>
> Across all experiments, we do not perform any instance-specific or object-specific hyperparameter tuning; instead, we simply adopt the **default or officially** recommended settings. This design choice enhances both the practicality and reproducibility of our method. Furthermore, the **experimental results in the ablation studies** demonstrate that our model can consistently produce plausible outcomes under these standard settings.
>
> The key hyperparameters used in our pipeline are listed below:
> | Multiview inpainting | Value |
> |-|-|
> | Inference timestep | 50 |
> | CFG | 5.0 |
>
> | Integration and enhanment | Value|
> |-|-|
> | Controlnet-tile Inference timestep | 32 |
> | Controlnet-tile CFG | 7.5 |
> | Controlnet-tile Strength | 0.25 |
>
> A **higher Strength** leads to **inconsistent or misaligned** results that do not match the preserved (visible) regions.
> In contrast, moderate Strength values (e.g., 0.25) reliably maintain alignment while enhancing detail quality.
>
> | Geometry and Texture Optimization | Value|
> |-|-|
> | LRM | official setting |
> | StableNormal | official setting |
>
> > W1:  2. Runtime
>
> Our model does not require substantial GPU resources. A single NVIDIA RTX 3090 GPU (24GB) is sufficient for running the entire pipeline efficiently.
> | Methods | Time |
> |-|-|
> | Inpainting | 5s |
> | Integration and enhanment | 13s|
> | Coarse mesh reconstruction|6s|
> | Geometry and texture refinement| 20s|
> | total | 44s |
>
> -----------------
>
> > W2. Comparison with Amodal3R.
>
> Thank you for your thoughtful suggestion.
> We report the performance of Amodal3R as follows. As shown in **Figure 11 of our paper**, we find that the results of Amodal3R often **misalign** the conditioned images and masks.
> Furthermore, we notice that the base model used by Amodal3R (Trellis) also faced **similar issues** with misalignment and inconsistencies, which in turn affected its ability to generate accurate completions.
>
> | Method | PSNR | LPIPS | CD | F-score |
> | - | - | - | - | - |
> | Amodal3R | 19.37 | 0.15 | 0.008 | 0.248 |
> | Ours | 23.35 | 0.09 | 0.005 | 0.389 |
>
> -----------
> > Q1. Large missing area
>
> Indeed, **large missing areas** are the main failure modes as reported in **Supplementary Materials A.5**. We also extend it to three datasets, including real-world and physically-simulated datasets in **A.1 of the main paper**.
>
> 1. Across all datasets, performance **consistently improves** as the **missing** region becomes **smaller**. This is expected, as **smaller missing** regions provide the model with **richer contextual information**, enabling more **accurate inference** of the missing shape.
>
> 2. Moreover, when the **missing** region becomes very **large**, the model inevitably gains **more flexibility** in generating **plausible content**. In such cases, the output may **differ** from the original **ground truth**, but this discrepancy should not necessarily be considered an “error.” This is because extremely **large missing regions** contain **little or no contextual guidance**, leading to inherently **ambiguous** reconstructions.
>
>
>
>
> ------------

---

> ### Author Response · Authors · 2025-11-28
>
> We truly appreciate your valuable feedback and the time you’ve dedicated to reviewing our manuscript. Your insights are incredibly helpful in refining our work, and we are grateful for your detailed suggestions.
>
> We would like to respectfully note that **many of the issues you raised** have **already** been **addressed** in the manuscript **prior to the rebuttal**, including:
> 1. **Large missing area** – Refer to **Supplementary Materials A.5**.
>
> Additionally, we would like to **highlight** that **several updates** have been **made** in the **new version of the manuscript**, particularly in the following areas:
>
> 1. **Robustness** – See **A.10** of the **main paper**.
> 2. **Runtime** – Updated in **A.7** of the **main paper**.
> 3. **Comparison with Amodal3R** – Discussed in **A.3** of the **main paper**.
> 4. **Large missing area** – More detailed analysis can be found in **A.1** of the **main paper**.
>
> We hope that these updates address your concerns, and we would be happy to provide further clarifications if needed. Your feedback is invaluable, and we look forward to any additional suggestions you may have.

---

### Official Review · Reviewer_5imQ · 2025-10-31

**Soundness:** 3
**Presentation:** 2
**Contribution:** 2
**Rating:** 4
**Confidence:** 3

**Summary:**

Authors propose a framework to restore geometry and texture from damaged 3D object. Specifically, authors use a multi-view model enhanced with a so-called mask-perceiver (which takes object incomplete mask) and depth-aware mask rectifier (which reconstructs occluded mask of interest). Afterwards, authors enhance multiview image from 256x256 and apply geometry-guided texture and shape optimization to refine the result.

**Strengths:**

1. Authors proposed a method which lets determine masks for multi-view inpainting, while other methods often require user-specified mask.
2. The design of mask self-perceiver and depth-aware mask rectifier is novel, to my knowledge.
3. Authors propose a novel way of generating synthetic dataset of broken objects.

**Weaknesses:**

1. Authors employ image enhancement and geometry + texture refinement as means to enhance overall quality of their pipeline. After careful review, I do not see any novelty in their proposed methodology, and consider it as test-time enhancement strategy as it has to be applied per object.
2. It is furthermore not clear whether the same texture / geometry optimization was used when inferring other methods.
3. Important baselines are missing (see question 3).
4. The quality of the output of mask self-perceiver is not convincing based on Fig 2. (see question 4).

**Questions:**

1. L088 "as simpler methods often fall short" - please elaborate.
2. Please provide more information on whether image enhancement or texture / geometry optimization technique was used when comparing with other methods, especially in table 1b.
3. Please add missing comparisons: MVInpainter in Table 1a, Amodal3R, Step1X or Hunyuan3D in Table 1b, ObjFiller3D in Table 2a.
4. Based on the visuals in Figure 2, the quality of the mask produced by the perceiver appears suboptimal. I therefore suggest that the authors (1) include additional visual examples and (2) consider performing an ablation study comparing their design with the straightforward DiffEdit approach (see Figure 2 in the main paper), which determines the masking region using, for instance, the difference between noise conditioned and unconditioned on text.

---

> ### Author Response · Authors · 2025-11-26
>
> We sincerely appreciate your recognition of **our contributions**, including enabling automatic mask determination for multi-view inpainting, the **novelty** of our mask self-perceiver and depth-aware rectifier, and our **novel** strategy for generating **synthetic** broken-object datasets. Your encouraging comments are truly valuable to us. Thank you for your valuable feedback!
>
> ---------
>
> > W1. Clarifying the Role and Novelty of Image Enhancement and Geometry/Texture Refinement.
>
> **• Image Integration and Enhancement.**
>
> We agree that **image enhancement itself** is **not** a **novel** idea. However, our **contribution** does not lie in simply applying enhancement, but in introducing an **auto-inferred mask** that **selectively guides enhancement** only on the **generated regions**. This **mask** serves as a **crucial linkage** between the first-stage multiview inpainting and the subsequent enhancement stage, ensuring that the **original content** is faithfully **preserved** while **refinement** is applied exclusively to the **synthesized areas**.
>
> **• Geometry and Texture Refinement.**
>
> We would like to further clarify that geometry and texture refinement are an **essential** component.
> **Without** this refinement, the **reconstructed object** is often **inconsistent with the input image** as shown in both quantitative and qualitative comparisons. These inconsistencies cause the **restored object** to deviate from its original appearance, making the final result **not faithful** to the input.
>
> > Q1. We elaborate on this in detail in our paper using blue text.
>
>
> ---------
>
> > W2 & Q2.
> 1. All the methods in Table 1b use our inpainted and enhanced images as inputs for a fair comparison.
> 2. Texture and geometry optimization are only applied to InstantMesh, which serves as our LRMs to infer coarse meshes.
> Hunyuan3D-2 can also be used as our LRMs. When applying texture and geometry optimization on it, the performance will be improved, as shown in the following table.
>
> | Method | PSNR | LPIPS | CD | F-score |
> | - | - | - | - | - |
> | Hunyuan3D-2 | 21.31 | 0.14 | 0.006 | 0.346 |
> | + optimization | 24.79 | 0.09 | 0.005 | 0.411 |
>
> ------
>
> > W3 & Q3. Missing comparisons
>
> **• MVInpainter.**
> We include results for MVInpainter. Similar to other baselines, it is **unable** to accurately perceive the regions that require inpainting. This limitation is reasonable, as MVInpainter is specifically designed for **object removal**, which is inherently **different** from our task. Object removal typically involves eliminating an **entire object**, whereas our task focuses on completing **partial regions of an object**, leading to fundamentally different requirements and challenges.
>
> | Method | PSNR | LPIPS | SSIM |
> | - | - | - | - |
> | MVInpainter | 11.12 | 0.29 | 0.79 |
> | Ours | **25.50** | **0.06** | **0.95** |
>
> **• Amodal3R.**
> We report the performance of Amodal3R as follows. As shown in **Figure 11 of our paper**, we find that the results of Amodal3R often **misalign** the conditioned images and masks.
> Furthermore, we notice that the base model used by Amodal3R (Trellis) also faced **similar issues** with misalignment and inconsistencies, which in turn affected its ability to generate accurate completions.
>
> **• Hunyuan3D-2.**
> The shape-generation and texture-generation models in Hunyuan3D-2 are very **large**, so we use the fast version for inference. Even with the fast version, generating meshes still takes a **long time**—for example, shape generation alone often requires several minutes, while the **texture-generation** pipeline typically takes **more than 30 minutes**.
> In contrast, our model produces a **fully textured mesh** in only about **one minute**. The **visual comparison** is shown in **A.5 of our paper**.
>
> | Method | PSNR | LPIPS | CD | F-score |
> | - | - | - | - | - |
> | Hunyuan3D-2 | 21.31 | 0.14 | 0.006 | 0.346 |
> | Amodal3R | 19.37 | 0.15 | 0.008 | 0.248 |
> | Ours | 23.35 | 0.09 | 0.005 | 0.389 |
>
> **• ObjFiller3D.**
> The author does not release their code, so we cannot compare our method with it.

---

> ### Author Response · Authors · 2025-11-26
>
> > W4 & Q4. 1. Additional Visual Examples.
>
> We provide more visual examples in Figure 12 of our paper.
>
> > W4 & Q4. 2. Comparison with DiffEdit.
>
> We report the mask IOU between the generated masks and the pseudo masks annotated by SAM2 as follows. The visual comparison is shown in **A.4 of our paper**. Our method significantly **outperforms** DiffEdit in terms of mask quality.
>
> DiffEdit relies **solely** on the difference between the noise conditioned and unconditioned on **text** to infer the mask. However, this approach is **suboptimal** because it does not take into account the **explicit image and depth information**, which are crucial for guiding the model to generate more accurate and contextually appropriate masks in the object restoration task. In contrast, our method incorporates both the image and depth as conditions, significantly improving the quality of mask generation.
>
> | Method | IOU |
> |- | -|
> | DiffEdit | 22.0 |
> | Ours | **49.9**|

---

> ### Author Response · Authors · 2025-11-28
>
> We truly appreciate your valuable feedback and the time you’ve dedicated to reviewing our manuscript. Your insights are incredibly helpful in refining our work, and we are grateful for your detailed suggestions.
>
> We would like to **highlight** that **several updates** have been **made** in the **new version of the manuscript**, particularly in the following areas:
>
> 1. **Q1**. We elaborate on this in detail in our paper using **blue text**.
> 2. **Comparison with MVInpainter** – See **A.9** of our **main paper**.
> 3. **Comparison with Amodal3R** – Refer to **A.3** of our **main paper**.
> 4. **Comparison with Huanyuan3D-2** – Discussed in **A.5** of our **main paper**.
> 5. **Additional Visual Examples & Comparison with DiffEdit** – Available in **A.4** of our **main paper**.
>
> We hope that these updates address your concerns, and we would be happy to provide further clarifications if needed. Your feedback is invaluable, and we look forward to any additional suggestions you may have.

---

### Official Review · Reviewer_1hjg · 2025-11-05

**Soundness:** 3
**Presentation:** 3
**Contribution:** 3
**Rating:** 4
**Confidence:** 3

**Summary:**

The paper introduces Restore3D, a system for joint restoration of geometry and texture for damaged objects from multi-view images. The pipeline first performs mask-free multi-view inpainting using a mask self-perceiver together with a depth-aware mask rectifier to decide what to keep versus synthesize, alleviating cross-view inconsistency and occlusion issues. The completed views are then integrated and enhanced before a coarse-to-fine 3D reconstruction stage: a large reconstruction model produces a coarse mesh, which is refined with normal priors for shape and UV back-projection with optimization for texture. Training pairs are generated automatically from G-Objaverse via Blender Boolean operations. Experiments on GSO, OmniObject3D, Objaverse, and out-of-domain case studies show consistent gains on PSNR/LPIPS/FID/SSIM for inpainting and CD/F-score for geometry, with ablations supporting the component choices and a reported ~20 s/object refinement time.

**Strengths:**

1. Originality: Tackling shape and texture restoration jointly from multi-view inputs—without user masks—is timely. The combination of a mask self-perceiver with a depth-aware rectifier is a sensible departure from single-view, mask-driven inpainting and geometry-only completion.
2. Design Rationality: The system is end-to-end and reasonably complete: synthetic data generation, mask-free inpainting, view integration/enhancement, and coarse-to-fine reconstruction with normal-prior refinement and UV-space texture optimization.
3. Broad Application: If the robustness holds, the approach is potentially useful for cultural-heritage restoration and occluded-object recovery, and it shows encouraging cross-domain generalization despite training primarily on synthetic data.

**Weaknesses:**

1 Data realism and domain gap: Breakage is synthesized via Boolean operations on G-Objaverse. This is practical but may not capture fracture statistics, material properties, or abrasion typical of real artifacts. The paper would benefit from a more systematic analysis of failure modes and long-tail categories.
2 Stacked dependencies and attribution: The method leans on several strong priors (Depth-Anything/StableNormal, Real-ESRGAN, ControlNet-Tile, LRMs). It is difficult to disentangle where the gains come from or to assess reproducibility under tighter compute budgets. More controlled diagnostics (e.g., removing Real-ESRGAN or swapping ControlNet-Tile) would clarify attribution.
3 Evaluation breadth: Beyond standard PSNR/FID/LPIPS/SSIM and CD/F-score, there is no assessment of texture/lighting fidelity from a perceptual standpoint, view-consistency scoring, or user preferences. Texture quality is not evaluated beyond SSIM or simple RGB losses.

**Questions:**

1. Data synthesis fidelity: How do Boolean-based fractures differ from real breakage (e.g., edge roughness, fragment size distribution), and does adding procedural noise narrow the gap? Any quantitative comparison on real fragments versus synthetic pairs?
2. Mask self-perceiver supervision: How are targets obtained for the rectified masks during training, and how often does the rectifier fail under depth errors? Diagnostics stratified by mask size and occlusion severity would be helpful.
3. Enhancement attribution: Could you ablate Real-ESRGAN versus ControlNet-Tile and the blending strategy, reporting view-consistency and texture sharpness before/after integration to pinpoint the source of gains?

---

> ### Author Response · Authors · 2025-11-26
>
> We sincerely appreciate your recognition of our method’s **originality, design rationality, and application potential**. Your positive comments on our joint shape–texture restoration, mask-free framework, and broad real-world relevance are truly encouraging. Thank you for your valuable feedback!
>
> > W1: 1. Data realism and domain gap
>
> We agree that our synthetic data may have a domain gap with real-world data. Therefore, **as shown in Figure 1 and Table 2 in the main paper**, we additionally evaluate our model on a **real-world dataset** and on a **physically simulated dataset**. In both settings, the model is trained **only** on our synthetic data and is tested **without any fine-tuning**. We find that our method still achieves **plausible and competitive** results on these **out-of-domain** datasets, which provides **empirical evidence** to show our model's **generalization** ability.
>
> > W1: 2. Analysis of failure modes
>
> Indeed, **large missing areas** are the main failure modes as reported in **Supplementary Materials A.5**. We also extend it to three datasets, including real-world and physically-simulated datasets, in **A.1 of the main paper**.
>
> 1. Across all datasets, performance **consistently improves** as the **missing** region becomes **smaller**. This is expected, as **smaller missing** regions provide the model with **richer contextual information**, enabling more **accurate inference** of the missing shape.
>
> 2. Moreover, when the **missing** region becomes very **large**, the model inevitably gains **more flexibility** in generating **plausible content**. In such cases, the output may **differ** from the original **ground truth**, but this discrepancy should not necessarily be considered an “error.” This is because extremely **large missing regions** contain **little or no contextual guidance**, leading to inherently **ambiguous** reconstructions.
>
> ------------------
>
> > W2: 1. Ablation studies for each component.
>
> We clarify the gains for each component as follows.
>
> **• Depth-Anything. & Q2: 2. Depth errors.** In the inpainting stage, our model uses depths to rectify masks. **As reported in Table 1(a) of the main paper**, we compare using **pseudo-depth** predicted by Depth-Anything with using **ground-truth depth**. Replacing ground-truth depth with Depth-Anything leads to only **a very small performance drop**. This shows that our method **does not** critically **depend on highly accurate depth** and that an off-the-shelf **depth estimator** is **sufficient** in practice, especially when ground-truth depth is unavailable.
>
> **• StableNormal.** It is only used to **refine the coarse meshes** inferred by LRMs. The gains from this step are **already reported in Table 3(b) of the main paper.**
>
> **• LRMs.** Without LRMs, a typical alternative is to start from a simple primitive (e.g., a sphere) and optimize its shape using our geometry losses.
> As reported in the following table, LRMs provide a much **better initialization**, leading to **faster convergence** and **improved reconstruction quality**.
> | method | CD | F-score |
> |-|-|-|
> |LRMs + geometry optimization| **0.005** | **0.389** |
> |Sphere + geometry optimization| 0.02 | 0.197 |
>
> > W2: 2. Compute budgets.
>
> Our model is computationally efficient, runs on modest GPU memory (single NVIDIA RTX 3090 GPU (24GB)), and delivers high-quality results.
> |Methods|	Time |
> |-|-|
> |Inpainting	| 5s |
> |Integration and enhanment|	13s |
> | Coarse mesh reconstruction | 	6s |
> | Geometry and texture refinement |	20s |
> | total	| 44s |
>
> -------

---

> ### Author Response · Authors · 2025-11-26
>
> > W3. Evaluation breadth.
>
> We **follow previous methods**, such as Instantmesh, Unique3d, and Trellis, to measure the texture quality of the reconstructed meshes using PSNR/LPIPS/FID/SSIM, and to measure geometry using CD and F-score.
>
> **• Texture/lighting fidelity.**
> 1. Our method is **not a physically based rendering (PBR)** approach. Appearance is represented in an implicit form where texture, material, and lighting are **entangled**. Therefore, we cannot report relighting performance.
>
> 2. However, our inpainting model can handle **different lighting settings** as shown in **Supplementary Materials A.4**. When we change the illumination in the input views, the inpainted appearances remain **stable and visually plausible**.
>
> **• View-consistency scoring.** We use MEt3R [1] to measure the multiview inpainted images. The table shows that our model is **better** than other methods and very **close to** the Ground Truth, further validating the **effectiveness** of our model.
>
> |Method|MEt3R|
> |-|-|
> |SD|0.44|
> |Controlnet|0.53|
> |Nerfiller|0.50|
> |Pix2gestalt|0.41|
> |Instant3dit|0.34|
> |Ours|**0.32**|
> |Ground Truth|**0.29**|
>
> [1] MEt3R: Measuring Multi-View Consistency in Generated Images. CVPR2025
>
> **• User preferences.**
> We provide user studies to measure the reconstructed meshes. 5 is the best score, 1 is the worst score. The results show our model outperforms other methods.
>
> |Method| geometry | texture |
> |-|-|-|
> |Open-LRM| 1.9| 2.1|
> |InstantMesh | 3.2 | 3.2|
> |Unique3D | 3.3| 3.5|
> |Direct3D| 3.0| - |
> |Trellis | 3.1| 3.0 |
> |Hunyuan3D| 3.5 | 3.6 |
> |Ours| 3.9 | 4.0|
>
>
> -----------
>
> > Q1. Boolean-based fractures vs. real breakage.
>
> **• Edge roughness.**
> 1. Pure **boolean operations** tend to produce **relatively smooth**, nearly planar fracture surfaces, while **real breakage** often exhibits **small-scale edge roughness** along the boundary.
> **However**, we argue that this difference in very fine edge roughness has **limited impact** on our model’s predictions as shown in **experiments** on unseen **real-world** scenarios and **physically simulated** case, which includes **unsmooth and irregular fractures**.
>
> 2. The **reason** is that our pipeline operates on rendered images at a resolution of 256×256, which are then passed through an image encoder that downsamples the feature maps to 32×32. Under this downsampling, high-frequency perturbations along the fracture boundary are effectively smoothed out. As a result, the model mainly relies on the coarse-scale geometry and global configuration of fragments (e.g., which parts are missing, how large the missing region is)
>
> **• Fragment size distribution.**
> 1. Our data synthesis pipeline **can generate different sizes of fragments**. Concretely, we use an icosphere or a cube with randomly sampled size and rotation angle, and randomly place this primitive inside the 3D bounding box of the original object. This procedure leads to a variety of fracture locations, resulting in different shapes of boundaries and different sizes of fragments.
> 2. We show the **Fragment-size proportion and performance** of our synthetic test dataset (GSO, Objaverse, OmniObject3D) and unseen dataset (Breaking Bad Dataset, Fantastic Breaks) in the **A.1 of our main paper**.
>
>
> ---------------
>
> > Q2: 1. Mask self-perceiver supervision.
>
> We only use the diffusion loss. The rectified masks are unsupervisedly learned from the model training.
>
> -----------

---

> ### Author Response · Authors · 2025-11-26
>
> > W2 & Q3. Image Integration and Enhancement.
>
> Thank you for your insightful suggestion.
> When **comparing our method with other inpainting methods**, we **do not use this pipeline** for a fair evaluation. The pipeline is **only used in the reconstruction stage** to provide **high-quality images** and to serve as the inputs for all methods being compared to ours.
>
> We conduct a more detailed ablation study as shown in the following Table. The **visualization results** are placed in the **A.2 of our paper**.
> We observed that:
> 1. Solely applying enhancement methods does not improve the quantitative metrics (as shown in Table b&c) but can improve visual quality.
> 2. The performance gains **mainly** originate from the **image integration**, which also **validates** that our rectified mask **well indicates** the regions requiring inpainting or preservation.
>
> **• Real-ESRGAN.**
> As mentioned in our paper, Real-ESRGAN is effective at preserving the patterns of low-resolution images with **minimal misalignment** and does **not introduce artifacts** that are inconsistent with the original style.
>
> **• ControlNet-Tile.**
> As mentioned in our paper, ControlNet-Tile offers advanced capabilities for enhancing image details, but it will **modify the original style** when a **high denoising step** is used.
> If used directly **without mask blending** (see Table f), it can **distort the integration** and alter the original pattern.
> **Mask blending** technique helps **preserve** the original patterns and **eliminate** gaps caused by image integration in the pixel space, **improving** the final results (see Table e).
>
> |Method (256px -> 1024px) | PSNR| LPIPS| SSIM|
> |-|-|-|-|
> |**a.** Baseline (Bilinear Upsampling)| 26.83	|0.10	|0.97|
> |**b.** 4x Real-ESRGAN | 26.59	|0.08	|0.97|
> |**c.** 4x Controlnet-tile| 26.56	|0.08	|0.96|
> |**d.** Real-ESRGAN + Image Integration| **27.13**	|**0.06**	| **0.97** |
> |**e.** Real-ESRGAN + Image Integration + Controlnet-tile (**used in our paper**) | 26.94	|**0.06**	|**0.97**|
> |**f.** Real-ESRGAN + Image Integration + Controlnet-tile (w/o mask blending)| 26.55|	0.07 |	**0.97**|
>
> **Overall**, the organization of this stage is **flexible**. The **key** ideas are:
>
> 1. Use ControlNet-Tile with a mask-blending strategy to eliminate color inconsistencies after image integration.
>
> 2. Upsample the image to the desired resolution using Real-ESRGAN, either before or after the integration step.
>
>
> ------------

---

> ### Author Response · Authors · 2025-11-28
>
> We truly appreciate your valuable feedback and the time you’ve dedicated to reviewing our manuscript. Your insights are incredibly helpful in refining our work, and we are grateful for your detailed suggestions.
>
> We would like to respectfully note that **many of the issues you raised** have **already** been **addressed** in the manuscript **prior to the rebuttal**, including:
>
> 1. **Data realism and domain gap** – **Figure 1** and **Table 2**.
> 2. **Analysis of failure modes** – **Supplementary Materials A.5** and **A.12**.
> 3. **Ablation studies for each component**:
>     - **Depth-Anything** & **Q2: Depth errors** – See **Table 1(a)**.
>     - **StableNormal** – Refer to **Table 3(b)**.
>     - **Visualization in Image Integration and Enhancement** – Available in **Figure 5**.
> 4. **Evaluation breadth**:
>     - **PSNR/LPIPS/FID/SSIM and CD/F-score** (aligned with previous methods).
>     - **Texture/lighting fidelity** – Discussed in **Supplementary Materials A.4**.
> 5. **Boolean-based fractures vs. real breakage**:
>     - **Edge roughness** – Refer to **Figure 1** and **Table 2**, Section 3.1.
>     - **Fragment-size performance** – Detailed in **Supplementary Materials A.5**.
>
> Additionally, we would like to **highlight** that **several updates** have been **made** in the **new version of the manuscript**, particularly in the following areas:
>
> 1. **Analysis of failure modes & Boolean-based fractures vs. real breakage** – More detailed analysis can be found in **A.1** of the main paper.
> 2. **The role of LRMs** – Updated in **A.6 of the main paper**.
> 3. **Compute budgets** – Discussed in **A.7 of the main paper**.
> 4. **View-consistency scoring & user preferences** – Expanded in **A.8 of the main paper**.
> 5. **Image Integration and Enhancement** – Additional analysis can be found in **A.2 of the main paper**.
>
> We hope that these updates address your concerns, and we would be happy to provide further clarifications if needed. Your feedback is invaluable, and we look forward to any additional suggestions you may have.

---

### Author Response · Authors · 2025-12-03

**Dear Area Chair,**

We sincerely thank you for handling our submission, and we are grateful to all reviewers for their valuable time and constructive feedback. Their comments have significantly improved our paper. Below, we summarize the key contributions and our responses to all reviewer concerns.

R1: 1hjg. R2: 5imQ. R3: aMfZ. R4: xzb5.       W: Weaknesses.  Q: Questions.

---

## **Summary of Contributions**

- **Technical Comprehensiveness and Systematic Design**: This work presents a **complete end-to-end system** that integrates a **data synthesis pipeline**, **mask-free multi-view inpainting**, **image integration and enhancement**, and a **coarse-to-fine reconstruction** process. The system **effectively supports** **3D object restoration** and demonstrates **promising results** (**Acknowledged by R1, R4**).

- **Novel, Well-Motivated Mask-Free Multi-View Restoration**: This work proposes a **novel**, **well-motivated**, and **original** approach to multi-view image restoration, **eliminating** the need for **manually defined masks**. **Experiments** validate the **effectiveness** of this approach (**Acknowledged by R1, R2, R3**).

- **Data Synthesis Pipeline**: This work introduces a novel **automated data synthesis pipeline** that generates paired incomplete and complete 3D objects, **addressing** the common challenge of **insufficient training data** for broken object restoration. The model **trained on synthetic data** demonstrates its **applicability** to both **real-world and physically simulated data**, confirming the **effectiveness** of the automated pipeline (**Acknowledged by R1, R2, R4**).

- **Experimental Results and Cross-Domain Generalization**: Experimental results show that the proposed method significantly **outperforms** existing baselines across **multiple datasets**, including both real and physically simulated broken objects. This approach holds great **potential** for applications in **cultural heritage restoration** and **occluded-object recovery** (**Acknowledged by R1, R3, R4**).

---

> ### Author Response · Authors · 2025-12-03
>
> ## **Summary of Revisions**
>
> We have carefully addressed **all concerns** raised by the reviewers:
>
> ---
>
> - **[Clarification]** We would like to **clarify** that these issues were **already** **addressed** in the **original manuscript**.
>
>   > [**W1.1 Data realism and domain gap**][R1] – We conduct cross-domain experiments as shown in Figure 1 and Table 2 of the main paper.
>
>   > [**Q1 Boolean-based fractures vs. real breakage**][R1]:
>     - Edge roughness – Refer to Figure 1 and Table 2, Section 3.1.
>     - Fragment-size vs. performance – Detailed in Supplementary Materials A.5.
>
>   > [**W1.2 Analysis of failure modes; Q1 Large missing area; W4 & Q4 Limitations**][R1, R3, R4] – Supplementary Materials A.5 and A.12.
>
>   > [**W3 Evaluation breadth**][R1]:
>     - PSNR/LPIPS/FID/SSIM and CD/F-score - Aligned with previous methods.
>     - Texture/lighting fidelity – Discussed in Supplementary Materials A.4.
>
>   > [**Ablation studies**]
>     - [Depth-Anything & **Q2.2** Depth errors][R1] – See Table 1(a) of the main paper (L389-390).
>     - [StableNormal][R1] – Refer to Table 3(b) of the main paper.
>     - [Q3; W3.2 & Q3.2 Visualization in Image Integration and Enhancement][R1, R4] – Available in Figure 5 of the main paper.
>     - [W3.1 & Q3.1 Masked Self-Perceiver][R4] – Table 3(a) and Figure 8.
>
>   > [**More baselines**]
>   - [W2 & Q2 Completion baselines (e.g., SD-Fusion)][R4] – Supplementary Materials, A.11.
>
>
> - **[Action]** We further **address raised issues** and **provide more details** in the **new revision**.
>
>   > [**W1.2 Analysis of failure modes & Q1 Boolean-based fractures vs. real breakage; Q1 Large missing area; W4 & Q4 Limitations**][R1, R3, R4] – More detailed analysis can be found in A.1 of the main paper.
>
>   > [**W1. Role and Novelty**][R2]:
>   - Image Integration and Enhancement:
>     1. Our contribution does not lie in **simply applying enhancement**, but in introducing an **auto-inferred mask** that **selectively guides enhancement** only on the generated regions.
>     2. This mask serves as a **crucial linkage** between the first-stage multi-view inpainting and the subsequent enhancement stage, ensuring that the **original content** is **faithfully preserved** while **refinement** is applied exclusively to the **synthesized areas**.
>
>   - Geometry and Texture Refinement:
>     1. Geometry and texture refinement are an **essential component**. **Without** this refinement, the reconstructed object is often **inconsistent** with the input image as **shown** in both **quantitative** and **qualitative** comparisons.
>     2. These **inconsistencies** cause the **restored object** to **deviate from** its **original appearance**, making the final result **not faithful** to the input.
>
>
>   > [**W3 Evaluation breadth**][R1]:
>     - View-consistency scoring & user preferences – Expanded in A.8 of the main paper.
>
>
>   > [**W2.2 Compute budgets; W1.2 Runtime**][R1, R3] – Discussed in A.7 of the main paper.
>
>   > [**W1.1 Robustness; Q3.3 Hyperparameters**][R3, R4] – See A.10 of the main paper.
>     - A small number of hyperparameters.
>     - Using official or regular settings.
>     - Without per-object hyperparameter tuning.
>
>   > [**Ablation studies**]
>     - [W2.1 The role of LRMs][R1] – Updated in A.6 of the main paper.
>     - [W2 & Q3; W3.2 & Q3.2 Image Integration and Enhancement][R1, R4] – Additional analysis can be found in A.2 of the main paper.
>
>   >[ **Writing**]
>     - [Q1 Drawback of simpler methods][R2]  - Blue text in Sec. 1 .
>     - [W2 & Q2 Comparison Details][R2] - Blue text in Sec. 4.1.
>
>   > [**More baselines**]
>     - [W3 & Q3 MVInpainter][R2] – See A.9 of our main paper.
>     - [W3 & Q3; W2 Amodal3R][R2, R3] – Refer to A.3 of our main paper.
>     - [W3 & Q3 Huanyuan3D-2][R2] – Discussed in A.5 of our main paper.
>     - [W3 & Q3 ObjFiller3D][R2] - The author does not release their code.
>     - [W4 & Q4 DiffEdit][R2] - Additional Visual Examples & Comparison with DiffEdit. Available in A.4 of our main paper.
>
>   > [**Technical Details**]
>    - [Q2.1 Mask self-perceiver supervision.][R1] The rectified masks are unsupervisedly learned from the model training.
>    - [W1 & Q1 3D consistency][R4] - A.11 of the main paper.
>
> ---

---

### Meta-Review · Area_Chair_DUZD · 2026-01-08

**Summary:**

This paper received mixed reviews. The main concerns raised in the reviews are:
1. potential gaps in synthetic and real fracture patterns (`1hjg`).
2. heavy reliance on large models (`1hjg`); the sequential components raise concerns over robustness (`aMfZ`, `xzb5`).
3. inadequate evaluation metrics (`1hjg`).
4. missing important baselines (`5imQ`, `aMfZ`, `xzb5`).
5. inadequate ablation studies (`xzb5`).
6. inadequate limitation discussion (`xzb5`).

Overall, this is an interesting submission and all reviewers find the results promising. However, the current evaluation remains insufficient even after the rebuttal. Several major concerns have not been fully addressed. I recommend Reject.

**Reviewer Concerns:**

1. Concern #1 is partially addressed by the argument that evaluations on real-world captured fractured artifact datasets presented in Fig. 1 and Tab. 2. However, only 6 real-world examples are presented and the metrics do not report geometric accuracy on real captured datasets.
2. Concern #2 is still outstanding. I do not find the corresponding revisions on limitation claimed in the rebuttal (A.5 and A.12). The rebuttal texts also do not sufficiently this concern.
3. Concern #3 is partially addressed by the new metrics using MEt3R and user studies.
4. Concern #4 is partially addressed by the new comparison results, but the extent of the comparison varies across baselines. For instance, only two examples are shown for comparison against Amodal3R.
5. Concern #5 is partially addressed by the new ablation results, but the performance difference is minimal.
6. Similar to Concern #2, I do not find the revision claimed in the text and cannot confirm whether concern has been addressed.

**Reviewer Scores:**

It is difficult to predict any of the reviewers would increase their rating as many of the concerns have not been fully addressed.

---

### Decision · Program_Chairs · 2026-01-26

Reject